# Controlling the Flow: Stability and Convergence for Stochastic Gradient Descent with Decaying Regularization

**Sebastian Kassing**[*]
Institute of Mathematics
Technical University of Berlin
10623 Berlin, Germany
kassing@math.tu-berlin.de

**Simon Weissmann**[*]
Institute of Mathematics
University of Mannheim
68159 Mannheim, Germany
simon.weissmann@uni-mannheim.de

**Leif Döring**
Institute of Mathematics
University of Mannheim
68159 Mannheim, Germany
leif.doering@uni-mannheim.de

## Abstract

The present article studies the minimization of convex, $L$-smooth functions defined on a separable real Hilbert space. We analyze regularized stochastic gradient descent (reg-SGD), a variant of stochastic gradient descent that uses a Tikhonov regularization with time-dependent, vanishing regularization parameter. We prove strong convergence of reg-SGD to the minimum-norm solution of the original problem without additional boundedness assumptions. Moreover, we quantify the rate of convergence and optimize the interplay between step-sizes and regularization decay. Our analysis reveals how vanishing Tikhonov regularization controls the flow of SGD and yields stable learning dynamics, offering new insights into the design of iterative algorithms for convex problems, including those that arise in ill-posed inverse problems. We validate our theoretical findings through numerical experiments on image reconstruction and ODE-based inverse problems.

## 1 Introduction

In this work we study the unconstrained optimization problem

$$\min_{x \in \mathcal{X}} f(x), \tag{1}$$

where $(\mathcal{X}, \langle \cdot, \cdot \rangle_{\mathcal{X}})$ is a separable real Hilbert space with inner product $\langle \cdot, \cdot \rangle_{\mathcal{X}}$ and induced norm $\|x\|_{\mathcal{X}}^2 = \langle x, x \rangle_{\mathcal{X}}$. The objective function $f : \mathcal{X} \to \mathbb{R}$ will be assumed to be differentiable, convex, and $L$-smooth with $\arg\min_{x \in \mathcal{X}} f(x) \neq \emptyset$. Moreover, we will always denote by $x_* \in \arg\min_{x \in \mathcal{X}} f(x)$ the minimum-norm solution, i.e. a minimum with $\|x_*\|_{\mathcal{X}} \leq \|\hat{x}\|_{\mathcal{X}}$ for all $\hat{x} \in \arg\min_{x \in \mathcal{X}} f(x)$. A common strategy for finding a point close to the minimum-norm solution is to employ regularization techniques. One popular approach from the optimization literature is to include Tikhonov regularization into (1) in the form of

$$\min_{x \in \mathcal{X}} f_\lambda(x), \quad f_\lambda(x) := f(x) + \frac{\lambda}{2} \|x\|_{\mathcal{X}}^2, \tag{2}$$

---

[*] These authors contributed equally to this work.

39th Conference on Neural Information Processing Systems (NeurIPS 2025).

where $\lambda \geq 0$ is called the regularization parameter. Since the regularized objective function $f_\lambda$ is $\lambda$-strongly convex for any $\lambda > 0$, there exists a unique minimum $x_\lambda = \arg\min_{x \in \mathcal{X}} f_\lambda(x)$ and many first order methods, such as stochastic gradient descent (SGD), are able to efficiently find $x_\lambda$.

Tikhonov regularization is a simple but effective method that appears in various contexts, such as statistics (e.g. ridge regression, [39]), classical inverse problems [31], including parameter estimation in partial differential equations [40] and image reconstruction [41, 18], dating all the way back to Tikhonov [70]. In the context of training neural networks, Tikhonov regularization is known under the name weight decay as the method decreases the norm of the neural network weights. One early reference is [50], for more recent work on the effect of weight decay on generalization we refer to [66], for LLM training to [22], and for a very recent experimental deep learning study to [29]. It is still a very much open problem to fully understand the different effects of weight decay, both from a practical but also the theoretical point of view in different optimization settings.

Recalling that $\|x_\lambda\|_\mathcal{X} \leq \|x_{\lambda'}\|_\mathcal{X} \leq \|x_*\|_\mathcal{X}$ for $\lambda' < \lambda$, see for instance [5], there is a trade-off between choosing $\lambda$ large and small. Large $\lambda$ speeds up convergence with the price of finding solutions that are too strongly regularized. On the other hand $\lim_{\lambda \to 0} \|x_\lambda - x_*\|_\mathcal{X} = 0$ suggests to turn down the regularization over time in order to ensure convergence to the minimum-norm solution. The present article provides a rigorous theoretical analysis for Tikhonov regularized stochastic gradient descent (reg-SGD) with decreasing (non-constant) regularization schedule $(\lambda_k)_{k \in \mathbb{N}_0}$. We show how to tune step-size and regularization schedules in order to achieve strong convergence to $x_*$. By strong convergence we refer to the convergence of the iterates $X_k$ in the sense $\lim_{k \to \infty} \|X_k - x_*\|_\mathcal{X} = 0$. For practical purposes, we derive how to optimally tune the decay rates of polynomial schedules.

## 1.1 Fixing the setup

Let us recall the classical Tikhonov regularized gradient descent scheme (reg-GD)

$$X_k = X_{k-1} - \alpha_k\big(\nabla f(X_{k-1}) + \lambda_k X_{k-1}\big), \tag{3}$$

which for constant $\lambda$ converges to $x_\lambda$ under suitable conditions on the step-size sequence $\alpha$. In many applications the gradient cannot be computed (or observed) exactly, instead only gradients with noisy perturbation are available. This leads to two equivalent formulations: one in which a noisy perturbation $D_k$ is added to the true gradient, and another in which the gradient is replaced by an estimated gradient $\widehat{\nabla f(X_{k-1})}$. These formulations are equivalent if we define the noise as $D_k = \widehat{\nabla f(X_{k-1})} - \nabla f(X_{k-1})$. We thus stick to the first setting but use the more accessible second notation for the pseudocode of Algorithm 1 below.

In this article, we study the regularized *stochastic* gradient descent scheme (reg-SGD) with *decreasing* regularization parameter $\lambda$. Let $(\mathcal{F}_k)_{k \in \mathbb{N}_0}$ be a filtration and $(X_k)_{k \in \mathbb{N}_0}$ be an adapted sequence defined recursively by

$$X_k = X_{k-1} - \alpha_k\big(\nabla f(X_{k-1}) + \lambda_k X_{k-1} + D_k\big), \tag{4}$$

where $\mathbb{E}[\|X_0\|_\mathcal{X}^2] < \infty$, $\alpha$ and $\lambda$ are sequences of (deterministic or random) non-negative reals, and $D := (D_k)_{k \in \mathbb{N}}$ is an adapted sequence of martingale differences, i.e. $\mathbb{E}[D_k \mid \mathcal{F}_{k-1}] = 0$ for all $k \in \mathbb{N}$. More precisely, in Theorem 2.1 we assume the sequences $\alpha := (\alpha_k)_{k \in \mathbb{N}}$ and $\lambda := (\lambda_k)_{k \in \mathbb{N}}$ to be predictable stochastic processes, i.e. $\alpha_k$ and $\lambda_k$ are $\mathcal{F}_{k-1}$-measurable for all $k \in \mathbb{N}$. The SGD formalism includes for instance stochastic gradients in finite-sum problems, where a random data point's gradient estimates the full gradient, see Example 1.3 below, and in expected risk minimization, where gradients are computed using samples from the data distribution.

---

**Algorithm 1** Regularized Stochastic Gradient Descent (reg-SGD)

---

**Require:** Initial guess $X_0$, number of iterations $N$, step-size schedule $\alpha$, regularization schedule $\lambda$
  1: **for** $k = 1$ to $N$ **do**
  2:    Compute unbiased gradient estimates: $\widehat{\nabla f(X_{k-1})} \approx \nabla f(X_{k-1})$.
  3:    Update parameters: $X_k = X_{k-1} - \alpha_k\big(\widehat{\nabla f(X_{k-1})} + \lambda_k X_{k-1}\big)$
  4: **end for**
  5: **return** $X_N$

---

We will further impose a second moment condition on the stochastic error terms $(D_k)_{k \in \mathbb{N}}$, which allows the noise term to grow with the optimality gap and the gradient norm. We emphasize that

throughout this work we will not impose any additional boundedness assumptions on the iterates of the reg-SGD scheme. Therefore, a priori the noise term might be unbounded. However, in the proofs below we show that, under weak assumptions on the step-size and regularization schedules, the additional regularization term implies almost sure boundedness of the iterates. This contrasts the dynamical behavior of standard SGD without regularization.

**Assumption 1.1.** *The objective function $f : \mathcal{X} \to \mathbb{R}$ is convex, continuously differentiable, and L-smooth. The latter means that $\nabla f : \mathcal{X} \to \mathcal{X}$ is globally L-Lipschitz continuous, i.e. there exists $L > 0$ such that $\|\nabla f(x) - \nabla f(y)\|_\mathcal{X} \leq L\|x - y\|_\mathcal{X}$ for all $x, y \in \mathcal{X}$. Furthermore, we assume that $\arg\min_{x \in \mathcal{X}} f(x) \neq \emptyset$ and denote by $x_* \in \arg\min_{x \in \mathcal{X}} f(x)$ the minimum-norm solution.*

For the noise sequence a typical ABC-type assumption is posed. The assumption is an important relaxation of bounded noise and can be verified in many applications [49, 36].

**Assumption 1.2.** *There exist constants $A, C \geq 0$ such that*

$$\mathbb{E}\big[\|D_k\|_\mathcal{X}^2 \mid \mathcal{F}_{k-1}\big] \leq A(f(X_{k-1}) - f(x_*)) + C, \quad k \in \mathbb{N}.$$

In contrast to the classical ABC condition, only two constants $A$ and $C$ appear. In Euclidean space when $f$ is differentiable, L-smooth, and bounded below, one has

$$\|\nabla f(x)\|^2 \leq 2L(f(x) - f(x_*)) \quad \text{for all } x \in \mathbb{R}^d, \tag{5}$$

see e.g. Lemma C.1 in [73]. The exact same argument (combining L-smoothness and the fundamental theorem of calculus) extends readily to the Hilbert space setting. Therefore, Assumption 1.2 is equivalent to the classical ABC-condition

$$\mathbb{E}\big[\|D_k\|_\mathcal{X}^2 \mid \mathcal{F}_{k-1}\big] \leq A(f(X_{k-1}) - f(x_*)) + B\|\nabla f(X_{k-1})\|_\mathcal{X}^2 + C, \quad k \in \mathbb{N},$$

for some $A, B, C \geq 0$.

**Example 1.3** (Mini-batch estimator for finite-sum problems)**.** *Consider the finite-sum optimization problem*

$$\min_{x \in \mathbb{R}^d} f(x) = \frac{1}{N} \sum_{i=1}^{N} f_i(x),$$

*where, for all $i = 1, \ldots, N$, $f_i : \mathbb{R}^d \to \mathbb{R}$ is convex and $L_i$-smooth. At iteration $k \in \mathbb{N}$, we can define a mini-batch estimator with mini-batch size $M \in \mathbb{N}$ via $g_k = \frac{1}{M} \sum_{i \in M} \nabla f_{I_{i,k}}(X_{k-1})$, where $(I_{i,k})_{i,k \in \mathbb{N}}$ is a family of iid. random variables that are uniformly distributed on $\{1, \ldots, N\}$. The corresponding gradient noise is defined as $D_k = \frac{1}{M} \sum_{i \in M}(\nabla f_{I_{i,k}}(X_{k-1}) - \nabla f(X_{k-1}))$ and satisfies*

$$\mathbb{E}\big[\|D_k\|^2 \mid \mathcal{F}_{k-1}\big] \leq \frac{4\bar{L}}{M}\big(f(X_{k-1}) - f(x_*)\big) + \frac{2\sigma_*^2}{M}, \tag{6}$$

*where $\bar{L} = \frac{1}{N} \sum_{i=1}^{N} L_i$ and $\sigma_*^2 = \frac{1}{N} \sum_{i=1}^{N} \|\nabla f_i(x_*)\|^2$. We will prove (6) in Lemma C.1 below.*

## 1.2 Contribution

The present article continuous a line of research on convergence properties for regularized differential equation based optimization flows (see e.g. [8, 7, 56] and the references therein). We show that the discretization of the stochastic differential equation setting considered in [56] yields a simple iterative scheme with similar convergence guarantees. It is non-trivial to establish a discrete iterative scheme that convergences fast to the minimum-norm solution $x_*$, as the step-size schedules $\alpha$ and regularization schedules $\lambda$ need to be balanced very carefully. In fact, while the assumptions we pose on the step-size schedule $\alpha$ are similar to the classical Robbins-Monro step-size conditions for convergence of SGD [63], the regularization schedule $\lambda$ has to satisfy two conflicting objectives. For a slowly decaying (almost constant) sequence $\lambda$, one can use the strong convexity of the regularized objective function $f_\lambda$ for all $\lambda > 0$ to show that $X_k$ is close to the minimum $x_{\lambda_k}$ of $f_{\lambda_k}$. However, this significantly slows down convergence to the minimum-norm solution due to the slow convergence of $\|x_{\lambda_k} - x_*\|_\mathcal{X}$. A crucial step in the analysis of reg-SGD will be to balance the two error terms

$$\|X_k - x_*\|_\mathcal{X} \leq \|X_k - x_{\lambda_k}\|_\mathcal{X} + \|x_{\lambda_k} - x_*\|_\mathcal{X}$$

appearing on the right-hand side. The main achievement of this article is to carry out last-iterate estimates that yield $L^2$ and almost sure convergence rates. In contrast to non-regularized SGD we obtain convergence to the minimum-norm solution (not just some solution) of the optimization problem while obtaining comparable rates for the optimality gap. The simulation in Figure 1 on the right shows the effect for $f(x_1, x_2) = (x_1 + x_2 - 1)^2$ with noisy gradients perturbed by independent Gaussians. While vanilla SGD converges to some minima (red dots), reg-SGD converges to the minimum-norm solution. Another important theoretical property that we reveal is that reg-SGD is more stable. It turns out that the iterative scheme is automatically bounded and cannot explode.

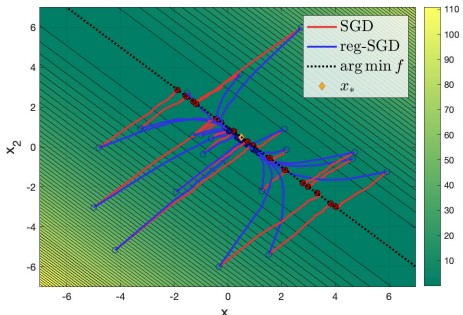

Figure 1: A comparison of SGD and reg-SGD, reg-SGD converges to $x_*$ for all initializations.

**Summary of main contributions:**

- $L^2$- and almost sure-convergence proof for last-iterates of reg-SGD to the minimum-norm solution under the ABC-condition *without* additional boundedness assumption on the stochastic iteration.

- $L^2$- and almost sure-convergence rates for polynomial step-size and regularization schedules.

- Experiments that show the stability of our polynomial step-size and regularization schedules.

### 1.3 Related work

To guide the reader we collect related articles and emphasize the line of research which we continue.

**Deterministic Tikhonov regularization:** The literature contains a number of articles on Tikhonov regularization with decreasing regularization parameter. For instance, in the context of deterministic optimization, this includes the analysis of first order ODEs [8, 26, 7] and second order ODEs [43, 6, 3, 7, 12]. Extensions to stochastic optimization in continuous time, in particular an analysis of the stochastic differential inclusion process, have been considered in [56]. Many statements are based on results for the solution curve $(x_\lambda)_{\lambda \geq 0}$ derived in [5]. More generally, differential inclusions for constrained convex optimization problems have been intensively studied in [9, 10, 11, 27, 60]. In Appendix D.2, we provide more details and illustrate the relation to Tikhonov regularization. Recently, the gradient flow for a fixed Tikhonov regularization has been analyzed in [21]. For small values of the regularization parameter $\lambda$, the optimization dynamics can be decomposed into two distinct phases: an initial fast convergence toward the set of minima, followed by a slow drift along this set that selects the minimizer with the smallest $\ell^2$-norm.

In this article, we extend methodologies for steepest descent flows to the stochastic discrete-time setting. Discrete time algorithms with decaying Tikhonov regularization have been analyzed in the context of iterative regularization schemes. For instance, in Chapter 5 of [14] iterative regularization is discussed to solve variational inequalities covering (3) for convex $f$ as a special case. Under certain conditions on the regularization decay and the step-sizes, strong convergence to the minimum-norm solution can be guaranteed. A related analysis has been considered for non-linear inverse problems [15]. In the specific application to inverse problems, (3) is also known as the modified Landweber iteration, where convergence is mainly studied for nonlinear forward models using a-priori and a-posteriori stopping rules [15, 47, 64]. The theoretical analysis is conducted in a non-convex setting and relies on the so-called tangential cone condition.

**Stochastic gradient descent (with regularization):** For recent results on convergence of SGD for possibly non-convex optimization landscapes we refer the reader to [57, 28, 72] and references therein. Note that due to the non-convexity, only convergence to a critical point can be shown without guarantees of optimality. In the smooth and convex case, almost sure convergence rates for the last-iterates of SGD without Tikhonov regularization under the ABC-condition for the noise have been derived in [51]. Therein, it was proved that for step-sizes $\alpha_k = C_\alpha k^{-\frac{2}{3} - \varepsilon}$ with $\varepsilon \in (0, \frac{1}{3})$ one has $f(X_k) - f(x_*) \in \mathcal{O}(k^{-\frac{1}{3} + \varepsilon})$ almost surely. [59] gives a rate of convergence for SGD with

polynomially decaying step-sizes in expectation. Their rate is optimal for the choice $\alpha_k = C_\alpha k^{-\frac{2}{3}}$ which yield the rate $\mathbb{E}[f(X_k) - f(x_*)] \in \mathcal{O}(k^{-1/3})$. Assuming uniform boundedness of the gradients and iterates, [67] increased this rate of convergence to $\mathcal{O}(k^{-1/2}\log(k))$ for the step-sizes $\alpha_k = C_\alpha k^{-\frac{1}{2}}$. These additional assumptions have been lifted in [52], where only bounded variance of the noise is assumed. Following [59, 52], the Ruppert-Polyak average also achieves the rate $\mathbb{E}[f(\bar{X}_k) - f(x_*)] \in \mathcal{O}(k^{-1/2}\log(k))$ for $\alpha_k = C_\alpha k^{-\frac{1}{2}}$. These results attain the lower bound for the optimization of $L$-smooth convex functions using first order algorithms that have access to unbiased gradient estimates with bounded variance derived in [2] up to a $\log(k)$ factor. More recently, almost sure convergence rates under a related setting have been derived in [65]. However, without any additional regularization one can only guarantee convergence towards some global minimum. The present article targets specifically algorithms that find the minimum-norm solution. We show that, when using reg-SGD one gets comparable convergence rates in the optimality gap as the ones for vanilla SGD cited above. Moreover, one can weaken the bounded variance assumption on the noise, while also achieving strong convergence to the minimum-norm solution. We also point to [38], where the role of regularization for the convergence of SGD for a prescribed number of optimization steps is discussed. Using a fixed regularization parameter, the authors derive a complexity bound for averaged SGD. However, they do not discuss the role of Tikhonov regularization for convergence towards a minimum-norm solution. Finally, in the context of inverse problems the regularization properties of (vanilla) SGD have been analyzed for linear [42, 44, 46] and non-linear forward models [45] based on a-priori and a-posteriori stopping rules. Moreover, in [33] SGD has been considered for inverse problems from a statistical point of view.

**Regularization effects in ML:** An exciting line of research that we do not touch directly is the explicit and implicit regularization effect of SGD appearing in ML training. We refer the reader for instance to the recent articles [61, 68, 16] and references therein. The relation to our work is that plain vanilla SGD tends to converge to minimum-norm solution in certain problems of practical relevance (in general it does not), while we prove that for convex problems Tikhonov regularized SGD with decaying regularization can always be made to converge to the minimum-norm solution.

## 2 Theoretical results

In this section, we present our main theoretical contributions concerning the convergence of stochastic gradient descent with decaying Tikhonov regularization. An abstract convergence result is presented, followed by quantitative rates of convergence for polynomial step-size and regularization schedules. For the ML practitioner, we derive optimal choices for the step-sizes and regularization parameters. All proofs are provided in Appendix D, where slightly more general statements are presented.

**Approach:** First, we carefully balance the step-size and the regularization parameters to ensure convergence of the energy function $E_k = f_{\lambda_{k+1}}(X_k) - f_{\lambda_{k+1}}(x_{\lambda_{k+1}})$. In Lemma B.3, we then obtain the estimate $\|X_k - x_{\lambda_k}\|_{\mathcal{X}} \le E_k/\lambda_k$, which links the distance to the regularized minimizer with the energy function. Since $\lambda_k$ also influences the decay of $E_k$, we must jointly control both quantities to ensure that $\lim_{k\to\infty} E_k/\lambda_k = 0$. Combined with the fact that $\lim_{\lambda\to 0} \|x_\lambda - x_*\|_{\mathcal{X}} = 0$, this yields strong convergence $\lim_{k\to\infty} \|X_k - x_*\|_{\mathcal{X}} = 0$ (both in $L^2$ and almost surely). Moreover, if a convergence rate for $\|x_\lambda - x_*\|_{\mathcal{X}}$ is known, the analysis allows us to also quantify a strong convergence rate for the iterates $X_k$ to $x_*$.

### 2.1 General convergence results

First, we present a general convergence statement for reg-SGD to the minimum-norm solution, both in the almost sure sense as well as the $L^2$-sense. The assumptions on the sequence of step-sizes $\alpha$ are similar to the Robbins-Monro step-size conditions. Regarding the sequence of regularization parameters $\lambda$, the assumptions for deriving almost sure convergence to the minimum-norm solution reflect the competing goals of using the strong convexity of $f_\lambda$ for $\lambda > 0$ and having sufficiently fast convergence of $x_\lambda \to x_*$. Compared to the almost sure convergence statement in Theorem 2.1, the second result, Theorem 2.2, establishes convergence in $L^2$ under arguably much weaker assumptions. In particular, the sequence $\lambda$ is allowed to decay at a very slow rate and no prior knowledge of the rate of convergence for $x_\lambda \to x_*$ is required.

We stress that we do not impose any boundedness assumptions of the reg-SGD scheme. In particular, the fact that $\sup_{k \in \mathbb{N}_0} X_k < \infty$ almost surely is a consequence of Theorem 2.1 which is guaranteed by the retracting force of the Tikhonov regularization.

**Theorem 2.1** (Almost sure convergence). *Suppose that Assumption 1.1 and Assumption 1.2 are fulfilled and let $(X_k)_{k \in \mathbb{N}_0}$ be generated by (4) with predictable (random) step-sizes and regularization parameters that are uniformly bounded from above. Moreover, we assume that almost surely the sequence $\lambda$ is decreasing to 0 and that*

$$\sum_{k \in \mathbb{N}} \alpha_k \lambda_k = \infty, \quad \sum_{k \in \mathbb{N}} \alpha_k^2 < \infty, \quad \text{and} \quad \sum_{k \in \mathbb{N}} \alpha_k \lambda_k \left( \|x_*\|_{\mathcal{X}}^2 - \|x_{\lambda_k}\|_{\mathcal{X}}^2 \right) < \infty. \qquad (7)$$

*Then $\lim_{k \to \infty} X_k = x_*$ almost surely.*

One can question how to verify the third assumption in (7) for practical applications. In Appendix E, we quantify the distance between $x_\lambda$ and $x_*$ in linear inverse problems satisfying a source condition, as well as in the situation, where $f$ satisfies a Łojasiewicz inequality. In general, one has no control for $\|x_*\|_{\mathcal{X}}^2 - \|x_\lambda\|_{\mathcal{X}}^2$, see [71]. We thus present a second result on $L^2$-convergence that holds also under a simpler condition. Here we require deterministic step-sizes and regularization parameters. Our requirements in (8) are very similar to the ones needed in the deterministic setting [14, Theorem 5.1 and Theorem 5.2] and are motivated by the corresponding deterministic result in continuous time [26, Theorem 2.2].

**Theorem 2.2** ($L^2$-convergence). *Suppose that Assumption 1.1 and Assumption 1.2 are fulfilled and let $(X_k)_{k \in \mathbb{N}_0}$ be generated by (4) with deterministic step-sizes and deterministic and decreasing regularization parameters $(\lambda_k)_{k \in \mathbb{N}}$. Moreover, assume that $\lambda_k \to 0$ and (7), or, alternatively, that*

$$\sum_{k \in \mathbb{N}} \alpha_k \lambda_k = \infty, \quad \alpha_k = o(\lambda_k), \quad \text{and} \quad \lambda_k - \lambda_{k-1} = o(\alpha_k \lambda_k). \qquad (8)$$

*Then $\lim_{k \to \infty} \mathbb{E}[\|X_k - x_*\|_{\mathcal{X}}^2] = 0$.*

In the next section, the theorems are made more explicit by choosing polynomial step-size and regularization schedules that allow us to derive convergence rates.

## 2.2 Convergence rates

We now go a step further and derive $L^2$- and almost sure-convergence rates for the particular choices of polynomial schedules

$$\alpha_k = C_\alpha k^{-q} \quad \text{and} \quad \lambda_k = C_\lambda k^{-p}, \quad p, q \in (0, 1).$$

Note that, due to Theorem 2.2, one has $\mathbb{E}[\|X_k - x_*\|_{\mathcal{X}}^2] \to 0$ if $q > p$ and $p + q < 1$. However, we can further derive the following convergence rates.

**Theorem 2.3** ($L^2$-rates for reg-SGD with polynomial schedules). *Suppose that Assumption 1.1 and Assumption 1.2 are satisfied. Let $C_\alpha, C_\lambda > 0$, $p \in (0, \frac{1}{2}]$ and $q \in (p, 1 - p)$. Let $(X_k)_{k \in \mathbb{N}_0}$ be generated by (4) with $\alpha_k = C_\alpha k^{-q}$ and $\lambda_k = C_\lambda k^{-p}$. If $q = 1 - p$ we additionally assume that $2C_\lambda C_\alpha > 1 - q$. Then it holds that $\lim_{k \to \infty} \mathbb{E}[\|X_k - x_*\|_{\mathcal{X}}^2] = 0$ and*

*(i)* $\mathbb{E}[f(X_k) - f(x_*)] \in \mathcal{O}(k^{-\min(p, q-p)})$,

*(ii)* $\mathbb{E}[\|X_k - x_{\lambda_{k+1}}\|_{\mathcal{X}}^2] \in \mathcal{O}(k^{-\min(1-q-p, q-2p)})$ *for $p \in (0, \frac{1}{3})$ and $q \in (2p, 1 - p)$.*

For a sequence of step-sizes $\alpha_k = C_\alpha k^{-q}$ with $q \in (0, \frac{2}{3}]$ one can set $\lambda_k = C_\lambda k^{-q/2}$ in order to get

$$\mathbb{E}[f(X_k) - f(x_*)] \in \mathcal{O}(k^{-\frac{q}{2}}).$$

For $q \in (\frac{2}{3}, 1)$ one can set $\lambda_k = C_\lambda k^{-1+q}$ in order to obtain

$$\mathbb{E}[f(X_k) - f(x_*)] \in \mathcal{O}(k^{-1+q}).$$

Therefore, we exactly recover the rates of convergence to some minimum for SGD without regularization derived in [59]. Recently, [52] improved the convergence rate for $q = \frac{1}{2}$ to

$\mathbb{E}[f(X_k) - f(x_*)] \in \mathcal{O}(k^{-1/2}\log(k))$. It is an interesting open question, whether the convergence rate of reg-SGD can be improved in this situation.

Finally, we derive almost sure convergence rates for regularized SGD. We highlight that Theorem 2.4 additionally gives almost sure convergence of reg-SGD to the minimum-norm solution for a specific choice of schedules without additional assumptions on the rate of convergence for $\|x_*\| - \|x_\lambda\|$.

**Theorem 2.4** (Almost sure-rates for reg-SGD with polynomial schedules). *Suppose that Assumption 1.1 and Assumption 1.2 are satisfied. Let $C_\alpha, C_\lambda > 0$, $p \in (0, \frac{1}{3})$ and $q \in (\frac{p+1}{2}, 1-p)$. Let $(X_k)_{k\in\mathbb{N}_0}$ be generated by (4) with $\alpha_k = C_\alpha k^{-q}$ and $\lambda_k = C_\lambda k^{-p}$. Then, it holds that $\lim_{k\to\infty} \|X_k - x_*\|_\mathcal{X} = 0$ almost surely and for any $\beta \in (0, 2q-1)$*

(i) $f(X_k) - f(x_*) \in \mathcal{O}(k^{-\min(\beta,p)})$ *almost surely,*

(ii) $\|X_k - x_{\lambda_{k+1}}\|_\mathcal{X} \in \mathcal{O}(k^{-\min(\beta-p,1-q-p)})$ *almost surely.*

For a sequence of step-sizes $\alpha_k = C_\alpha k^{-q}$ with $q \in (\frac{2}{3}, 1)$ one can set $\lambda_k = C_\lambda k^{-1+q+\varepsilon}$ with $0 < \varepsilon < 1-q$ to get almost surely

$$f(X_k) - f(x_*) \in \mathcal{O}(k^{-1+q+\varepsilon}),$$

which is the vanilla SGD rate of convergence to some minimum that has been recently derived in [51].

**Remark 2.5.** *In Theorem D.5 of the appendix we also provide a theorem on convergence rates for deterministic reg-GD (3) with polynomial step-size and regularization schedules.*

**Summary:** Incorporating carefully chosen vanishing Tikhonov regularization helps mitigate an exploding optimization sequence (the process $(X_k)_{k\in\mathbb{N}_0}$ is always bounded without further assumptions), ensures convergence to the minimum-norm solution, and achieves convergence rates in the optimality gap comparable to those of plain vanilla SGD.

## 2.3 Refinements under Łojasiewicz condition

In this final result section, we refine the above results under stronger assumptions on $f$. We use ideas that were recently used for continuous-time optimization schemes, see [56]. Let us assume $f$ satisfies the Łojasiewicz condition

$$(f(x) - f(x_*))^\tau \leq C\|\nabla f(x)\|_\mathcal{X} \quad \text{for all } x \in f^{-1}([f(x_*), f(x_*) + r]). \tag{9}$$

for some $C, r > 0$ and $\tau \in [0, 1)$. It then follows (and this is what we actually need) that there exist $C_{\text{reg}} > 0$ such that

$$\|x_\lambda - x_*\|_\mathcal{X} \leq C_{\text{reg}}\lambda^\xi, \quad \lambda \in (0, 1], \tag{10}$$

with $\xi = \frac{1-\tau}{2}$, see [56]. We provide further discussion in Appendix E. Note that (9) is sufficient to guarantee (10), however, in the subsequent convergence rates we rely only on (10). Now we use that

$$\|X_k - x_*\|_\mathcal{X}^2 \leq 2\|X_k - x_{\lambda_{k+1}}\|_\mathcal{X}^2 + 2\|x_{\lambda_{k+1}} - x_*\|_\mathcal{X}^2$$

so that we can bound the distance to the minimum-norm solution by (10) and the statements derived in Theorem 2.3 and Theorem 2.4.

Regarding the convergence in $L^2$, Theorem 2.3 together with (10) implies the following strong convergence rates in $L^2$:

**Corollary 2.6** (Strong $L^2$ convergence rates). *Suppose that the conditions of Theorem 2.3 are satisfied and assume that (10) is in place for some $\xi > 0$. Then it holds that*

$$\mathbb{E}[\|X_k - x_*\|_\mathcal{X}^2] = \mathcal{O}(k^{-\min(1-q-p,q-2p,2\xi p)}).$$

Thus, we get the optimal rate of convergence for $p = \frac{1}{4\xi+3}$ and $q = \frac{1+p}{2}$, which gives

$$\mathbb{E}[\|X_k - x_*\|_\mathcal{X}^2] = \mathcal{O}(k^{-\frac{2\xi}{4\xi+3}}).$$

For almost sure convergence, Theorem 2.4 together with (10) implies strong a.s. convergence rates:

**Corollary 2.7** (Strong a.s. convergence rates). *Suppose that the conditions of Theorem 2.4 are satisfied and assume that* (10) *is in place for some $\xi > 0$. Then for all $\beta \in (0, 2q-1)$ it holds that*

$$\|X_k - x_*\|_{\mathcal{X}}^2 = \mathcal{O}(k^{-\min(1-q-p,\beta-p,2\xi p)}) \quad \text{almost surely.}$$

Let $\varepsilon > 0$ and choose $\beta = 2q - 1 - \varepsilon$. Then, for the optimal values $p = \frac{1}{6\xi+3}$ and $q = \frac{2}{3}$ we get

$$\|X_k - x_*\|_{\mathcal{X}}^2 = \mathcal{O}(k^{-\frac{2\xi}{6\xi+3}-\varepsilon}), \quad \text{almost surely.}$$

In Figure 2, we illustrate the convergence rate of $\|X_k - x_*\|_{\mathcal{X}}^2$ depending on the decay-rates $p, q$ of schedules $\alpha$ and $\lambda$ in the situation where $f$ satisfies a Polyak-Łojasiewicz inequality, i.e. (9) is satisfied with $\tau = \frac{1}{2}$ and, thus, (10) is satisfied with $\xi = \frac{1}{4}$. In Appendix A.2 we provide a numerical experiment studying the behavior of convergence when implementing reg-SGD for different choices of $\alpha$ and $\lambda$.

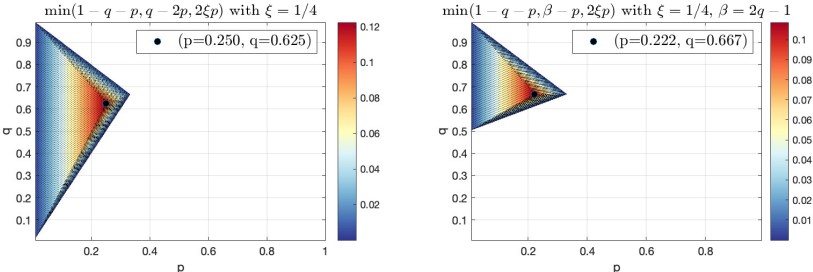

Figure 2: Optimal choices of $p$ and $q$. Left: convergence rate for $\mathbb{E}[\|X_k - x_*\|_{\mathcal{X}}^2]$ in the situation of Corollary 2.6. Right: almost sure convergence rate for $\|X_k - x_*\|_{\mathcal{X}}^2$ in the situation of Corollary 2.7 under the Polyak-Łojasiewicz inequality.

## 3 Practical implications

In this section, we discuss the relevance and application of reg-SGD with fine-tuned step-size and regularization schedules in the particular setting of linear inverse problems. We perform a concrete experiment to confirm on image reconstruction of tomography images the strength of our theoretically derived step-size and regularization schedules.

### 3.1 Why is reg-SGD important?

As a motivation, we consider a classical linear inverse problem posed in a Hilbert space [17, 31]. Let $\mathcal{X}$ and $\mathcal{Y}$ be two (separable) Hilbert spaces, and let $A : \mathcal{X} \to \mathcal{Y}$ be a bounded linear operator. Given the observation $y \in \mathcal{Y}$ the task of the inverse problem is to reconstruct $x \in \mathcal{X}$ such that $Ax = y$. The reconstruction problem is in general ill-posed, since the solution $Ax = y$ is typically non-unique. In particular, when $A$ has a non-trivial null space, there exist infinitely many solutions. Moreover, when $A$ is a compact operator the generalized Moore-Penrose inverse $A^\dagger$ is unbounded. As a consequence small perturbations in the data can lead to large variations in the reconstruction. One popular approach to solving the inverse problem is to select a stable reconstruction based on the minimum-norm solution

$$x_* := \arg\min\left\{ \|\hat{x}\|_{\mathcal{X}} \,\Big|\, \hat{x} \in \arg\min_{x \in \mathcal{X}} \|Ax - y\|_{\mathcal{Y}} \right\}.$$

Finding minimum-norm solutions is, as we also show in the present article, closely related to reg-SGD. When the observation $y$ is in the range of $A$, then the unique minimum-norm solution is given by $x_* = A^\dagger y$. In practice, the data space $\mathcal{Y}$ is often described as a function space of variables $s \in D \subset \mathbb{R}^d$ to $\mathbb{R}$ (e.g., in integral equations or tomography), where $s$ may model a sensor location or angle. Hence, the inversion can be formulated as a risk minimization problem involving data samples

$$y_i = A[x^\circ](s_i) + \sigma\epsilon_i \in \mathbb{R}, \quad i = 1, \dots, n,$$

generated by observations of some forward-mapped ground truth $x^\circ \in \mathcal{X}$ perturbed by noise $\epsilon_i$. The empirical objective can be formulated as

$$\min_{x \in \mathcal{X}} f(x), \quad f(x) := \frac{1}{n} \sum_{i=1}^{n} \left| A[x](s_i) - y_i \right|^2.$$

In our analysis we assume access to unbiased gradient estimators for noise-free data $y_i$ in the finite data regime, or for noisy data in the infinite data regime ($n \to \infty$). When analyzing finite noisy data, it is typically necessary to incorporate additional regularization, such as early stopping based on Morozov's discrepancy principle [4, 20, 58]. In practical applications, first-order optimization methods, and in particular the use of reg-SGD, is gaining popularity as an efficient approach for solving large-scale inverse problems [30, 24]. It would be interesting to explore whether our analysis can be extended to more advanced variational regularization schemes on constrained or non-smooth optimization problems [25].

In what follows, we present results from an experiment on a task of image reconstruction based on the Radon transformation. This experiment demonstrates the relevance of carefully tuning decreasing step-size and regularization schedules. Two additional experiments are provided in Appendix A that highlight the performance of our theoretically derived optimal schedules.

### 3.2 Fine-tuned reg-SGD for X-ray tomography

In the context of X-ray tomography, the Radon transform models how a two-dimensional image $x(z_1, z_2)$ is mapped to its projection data $R_\theta[x](\cdot)$ via line integrals along rays oriented at various angles $\theta \in [0, \pi)$, see e.g. [37] for details. These projections are obtained by integrating the image along parallel lines, simulating the physical process of X-ray attenuation. Formally, the forward Radon transform at angle $\theta$ is defined as

$$f \mapsto R_\theta[x](t) = \int_{\mathbb{R}} x(t\cos(\theta) - s\sin(\theta), t\sin(\theta) + s\cos(\theta)) \, ds$$

where $x(z_1, z_2)$ is the image to be reconstructed, $t \in \mathbb{R}$ denotes the location along the detector array orthogonal to the projection direction direction $\theta$. For numerical implementation, the Radon transform is discretized over a grid of pixels and a finite set of lines and projection angles. The inverse problem then consists of the reconstruction of an unknown image $x^\dagger$ from its noisy or incomplete measured projection data $R_\theta[x]$. For instance, the Radon transform may model X-rays passing through an object, and the reconstruction corresponds to inferring the internal structure of this object from these measurements, similar to assembling a complete image from multiple shadow-like projections. We formulate the reconstruction as the optimization problem

$$\min_x \sum_{i=1}^{M} \frac{1}{2} \|\mathcal{R}_{\theta_i}[x] - g_{\theta_i}\|^2.$$

We carried out an experiment, reconstructing an image from it's Radon transform (see Figure 3) solving the ill-posed optimization problem using SGD and reg-SGD with our optimal step-size schedule and a more aggressive regularization schedule. All details of the implementation are provided in Appendix A.1. The experiment demonstrates the strength of our fine-tuned step-size and regularization schedules. While our optimal schedules ($p = \frac{1}{3}$, $q = \frac{2}{3}$) yield fast convergence to the minimum-norm solution, a more aggressive schedule ($p = q = \frac{2}{3}$) stagnates at a suboptimal level. More critically, vanilla SGD with theoretically optimal step-sizes even fails to produce feasible reconstructions. To illustrate this, in Figure 4 we compare the reconstructed images from reg-SGD with the optimal rates from our analysis, reg-SGD with more aggressive rates, vanilla SGD, and the minimum-norm solution $x_*$, which is computed via the Moore-Penrose pseudoinverse $x_* = A^\dagger y$. Additionally, we plot both the expected and a.s. optimality gap in Figure 5 as well as the $L^2$- and pathwise-error to the minimum-norm solution Figure 5. In this experiment, SGD shows faster convergence in terms of the optimality gap, but ultimately fails to converge to the minimum-norm solution.

## 4 Conclusion and future work

We analyzed convergence properties of SGD with decreasing Tikhonov regularization. For convex optimization problems that may have infinitely many solutions, we showed that the regularization

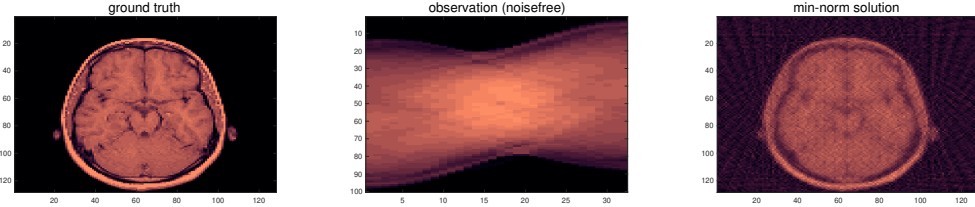

Figure 3: Left: base image. Middle: Radon transform. Right: minimum-norm solution $x_*$.

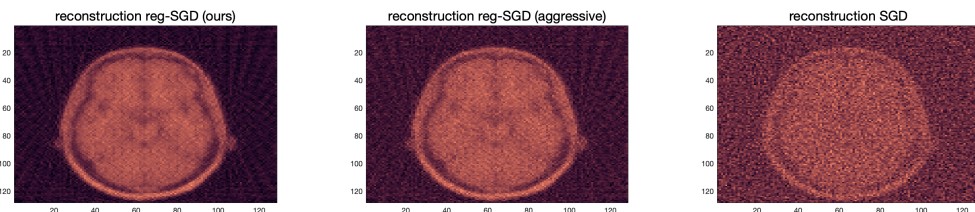

Figure 4: Left: reconstruction using reg-SGD with our optimal schedules. Middle: reconstruction using reg-SGD with more aggressive schedules. Right: reconstruction using vanilla SGD.

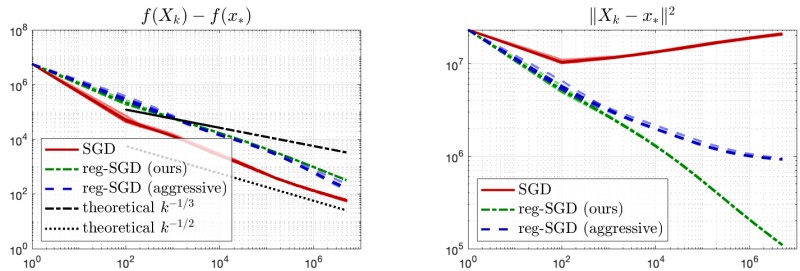

Figure 5: Left: pathwise optimality gap $f(X_k) - f(x_*)$. Right: pathwise squared error to the minimum-norm solution $\|X_k - x_*\|^2$. Each curve represents one of 10 independent runs, each of length $N = 5 \cdot 10^6$. The red shaded lines depict individual runs of SGD, while the green dash-dotted and blue dashed shaded lines correspond to reg-SGD. The red solid line shows the average error across runs for SGD, the green bold dash-dotted and blue dashed line shows the average for reg-SGD, and the black dashed line indicates the theoretical convergence rate.

can always be chosen to guarantee convergence (almost surely and in $L^2$) to the minimum-norm solution. In fact, we provided guidance on explicit choices for polynomial step-size and regularization schedules that ensure best (in the sense of our upper bounds) convergence rates. On the way we revealed interesting mathematical insight into the effect of regularization. In contrast to plain vanilla SGD, boundedness of the approximation sequence is always ensured. A number of concrete applications was provided to show that our theoretical best schedules indeed are consistent with experimental observations, specifically in the experiments of Appendix A.2. Since our analysis is limited to the smooth convex setting without constraints, for future work it could be interesting to

- extend results beyond the convexity assumption on $f$, e.g. using gradient domination properties [32] or the tangential cone condition which is commonly employed in iterative regularization methods for non-linear inverse problems [47],

- experiment with our suggested decreasing regularization in deep learning problems,

- use decreasing regularization schedules to better understand the relation of implicit and explicit regularization present in SGD, and

- study other regularization variants in situations in which minimum-norm solutions are not desirable (e.g. linear inverse problems with noisy data).

## Acknowledgments

We thank the reviewers and the area chair for their valuable feedback. Our sincere thanks to Adrian Riekert, whose discussions and feedback were instrumental in shaping the project from its outset. SK acknowledges funding by the Deutsche Forschungsgemeinschaft (DFG, German Research Foundation) – CRC/TRR 388 "Rough Analysis, Stochastic Dynamics and Related Fields" – Project ID 516748464.

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

# A    Omitted details and additional numerical experiments

In the following section, we give a detailed description of the implementation for our numerical experiment conducted in Section 3. Moreover, we provide two additional experiments.

## A.1    Implementation details of the Radon transform

In the example of Section 3.2, we discretize the Radon transform using a fixed set of 32 equally spaced projection angles $\theta_i \in [0, \pi)$, $i = 1, \ldots, 32$ and use 100 parallel rays per angle. The unknown image is defined on a $128 \times 128$ pixel grid and represented as a vector $x^\dagger \in \mathbb{R}^d \cong \mathbb{R}^{128 \times 128}$, $d = 128^2$. Given any image $x \in \mathbb{R}^d$ its discretized Radon transform is implemented as a matrix-vector product $Ax$, where $A \in \mathbb{R}^{K \times d}$ is the forward operator, and $K = 32 \times 100$, corresponds to the total number of measurements (i.e., the number of angle-ray combinations). Each row of $A$ represents a discrete line integral along one ray at a given projection angle. The objective function is then defined as

$$f(x) := \frac{1}{2} \|Ax - g\|^2, \quad x \in \mathbb{R}^d,$$

where $g = (g_{\theta_1}, \ldots, g_{\theta_{32}}) \in \mathbb{R}^K$ collects all projection measurements.

We implemented both SGD and reg-SGD by partitioning the forward operator $A \in \mathbb{R}^{K \times d}$ into blocks $A_i \in \mathbb{R}^{100 \times d}$, each corresponding to a fixed projection angle $\theta_i$, $i = 1, \ldots, 32$. At each iteration, the angle $\theta_i$ is sampled uniformly at random, and the gradient of $f$ is approximated by $\nabla f_i(x) = A_i^\top (A_i x - g_{\theta_i}) \in \mathbb{R}^d$ and additionally perturbed by independent noise following a multivariate normal distribution with zero mean and covariance $0.5^2 \cdot \mathrm{Id}$. For SGD we chose the step-size schedule $\alpha_k = 20k^{-1/2}$. For reg-SGD we chose $\alpha_k = 20k^{-2/3}$ and regularization $\lambda_k = 0.01k^{-1/3}$. Moreover, we compare to reg-SGD with $\alpha_k = 20k^{-2/3}$ and regularization $\lambda_k = 0.01k^{-2/3}$, i.e., reg-SGD with a too fast decay of regularization. We initialize all algorithms for each repetition at zero.

## A.2    A toy example

In this section we present a didactic toy example from [7], where the regularization error in terms of $\|x_\lambda - x_*\|_{\mathcal{X}}$ can be calculated exactly. Consider the objective function

$$f(x_1, x_2) := \frac{1}{2}(x_1 + x_2 - 1)^2$$

with unique minimum-norm solution $x_* = (1/2, 1/2)$, see the plot in Section 1.2. Note that there exist infinitely many global minima of $f$. Incorporating Tikhonov regularization results in

$$f_\lambda(x_1, x_2) = f(x_1, x_2) + \frac{\lambda}{2}(x_1^2 + x_2^2) \quad \text{with} \quad x_\lambda = \left( \frac{1}{2 + \lambda}, \frac{1}{2 + \lambda} \right).$$

such that the residuals in the Euclidean distance of $\mathbb{R}^2$ are bounded by

$$\|x_* - x_\lambda\| = \frac{\lambda}{\sqrt{2}(2 + \lambda)} \leq \frac{\lambda}{2\sqrt{2}}.$$

Therefore, equation (10) is satisfied with $\xi = 1$.

*Implementation details:* We have implemented both vanilla SGD and reg-SGD by hand and initialized both algorithms with same initial state $X_0 \sim \mathcal{N}(0, 1)$ and perturbed the exact gradient $\nabla f$ in each iteration by independent noise following a multivariate normal distribution with zero mean and covariance $0.1^2 \cdot \mathrm{Id}$. For SGD we chose the step-size schedule $\alpha_k = 0.1k^{-1/2}$, $k \in \mathbb{N}$. For reg-SGD we chose $\alpha_k = 0.1k^{-q}$ and regularization $\lambda_k = k^{-p}$, where $p = \frac{1}{4\xi + 3}$, $q = (1 + p)/2$ when considering the $L^2$ convergence rates and $p = (6\xi + 3)^{-1}$, $q = 2/3$ when considering the almost sure convergence rates see Corollary 2.6 and Corollary 2.7.

The plots of Figures 6 and 7 illustrate that reg-SGD converges to the minimum-norm solution both in $L^2$ (Figure 6) and almost surely (Figure 7), as indicated by the vanishing squared error. In contrast, SGD does not converge to the minimum-norm solution, although it achieves convergence in the expected (Figure 6) and pathwise optimality gap (Figure 7). This highlights the regularization effect

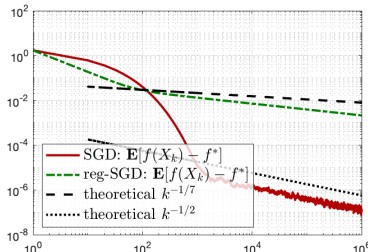 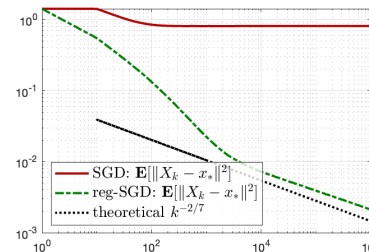

Figure 6: Left: expected optimality gap $\mathbb{E}[f(X_k) - f(x_*)]$. Right: $L^2$-error to the minimum-norm solution $\mathbb{E}[\|X_k - x_*\|^2]$. Each curve is computed over 100 independent runs of length $N = 10^6$. The red line shows the average performance of SGD, the green line represents reg-SGD, and the black dotted lines indicate the corresponding theoretical convergence rates from our theorems.

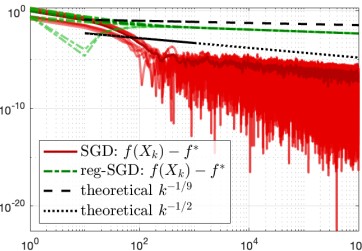 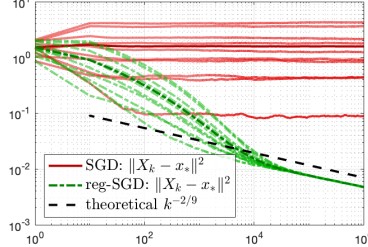

Figure 7: Left: pathwise optimality gap $f(X_k) - f(x_*)$. Right: pathwise squared error to the minimum-norm solution $\|X_k - x_*\|^2$. Each curve represents one of 10 independent runs, each of length $N = 10^6$. The red shaded lines depict individual runs of SGD, while the green dash-dotted shaded lines correspond to reg-SGD. The red solid line shows the average error across runs for SGD, the green bold dash-dotted line shows the average for reg-SGD, and the black dashed line indicates the theoretical convergence rate.

of reg-SGD in guiding the iterates toward the unique minimum-norm solution as also indicated in Figure 1.

Next, we compare different choices of $(p, q)$ for reg-SGD. In particular, we run reg-SGD with $\alpha_k = 0.2qk^{-1/2}$ and $\lambda_k = k^{-p}$ for the choices

$$(p, q) \in \{(0.111, 0.667), (0, 0.667), (0.67, 0.5), (0.111, 0.29)\}.$$

Moreover, we increase the noise covariance to $\mathcal{N}(0, \mathrm{Id})$. The expected convergence behavior is shown in Figure 8 while the resulting errors are displayed in Figure 9. As expected, we do not observe convergence for the choices $(0, 0.667)$ and $(0.67, 0.5)$ as the regularization is not turned off, respectively turned off too fast. We observe convergence both a.s. and in $L^2$ when choosing

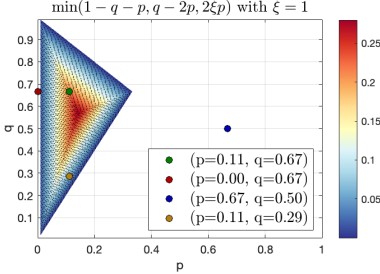 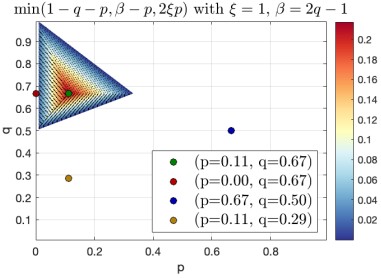

Figure 8: Convergence rate for $\mathbb{E}[\|X_k - x_*\|_{\mathcal{X}}^2]$ in the situation of Corollary 2.6 (left) and almost sure convergence for $\|X_k - x_*\|_{\mathcal{X}}^2$ in the situation of Corollary 2.7 in the considered setting of Appendix A.2 with $\xi = 1$. Furthermore, we display the choices $(p, q) \in \{(0.111, 0.667), (0, 0.667), (0.67, 0.5), (0.111, 0.29)\}$ which are simulated and displayed in Figure 9.

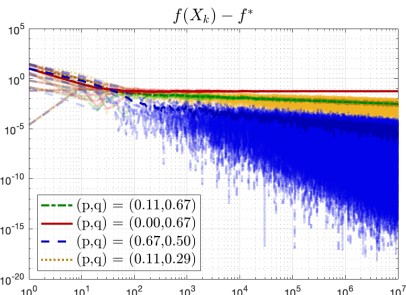
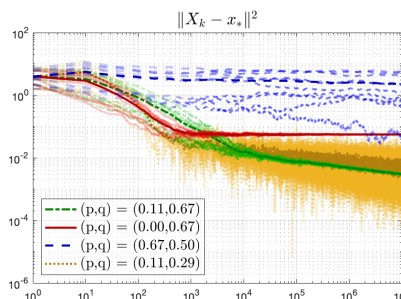

Figure 9: Left: pathwise optimality gap $f(X_k) - f(x_*)$. Right: pathwise squared error to the minimum-norm solution $\|X_k - x_*\|^2$. Each curve represents one of 10 independent runs, each of length $N = 10^7$. The shaded lines depict individual runs of reg-SGD. The solid lines show the average errors for reg-SGD. The different colors correspond to various choices of $(p, q) \in \{(0.111, 0.667), (0, 0.667), (0.67, 0.5), (0.111, 0.29)\}$.

$(0.111, 0.667)$ as suggested by our theory. In contrast, when choosing $(0.111, 0.29)$ our theoretical results suggest that the step-size decay is too slow, which we observe in a high variance of the deviation to the minimum-norm solution. In the final experiment, we examine the effect of the initial value $\alpha_1 > 0$ in the step-size schedule. For SGD, we set $\alpha_k = \alpha_1 k^{-1/2}$, while for reg-SGD we fix $\lambda_k = k^{-0.111}$ and use the step-size schedule $\alpha_k = \alpha_1 k^{-0.667}$. We report both the pathwise optimality gap and the pathwise squared error to the minimum-norm solution for SGD (Figure 10) and reg-SGD (Figure 11) under varying initial step sizes $\alpha_1 \in \{0.01, 0.1, 1, 2\}$.

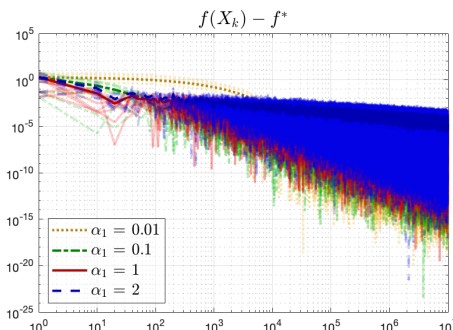
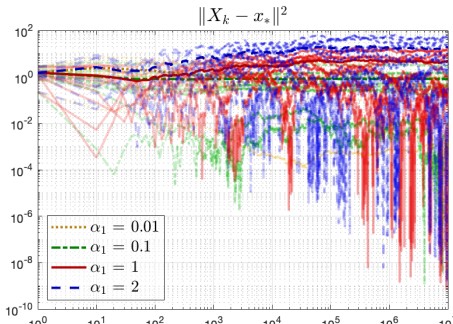

Figure 10: Left: pathwise optimality gap $f(X_k) - f(x_*)$. Right: pathwise squared error to the minimum-norm solution $\|X_k - x_*\|^2$. Each curve represents one of 10 independent runs of SGD, each of length $N = 10^7$. The shaded lines depict individual runs of SGD. The solid lines show the average errors for SGD. The different colors correspond to various choices of $\alpha_1 \in \{0.01, 0.1, 1, 2\}$.

### A.3 ODE based inverse problem.

In the following example, we consider a linear inverse problem arising from the one-dimensional elliptic boundary value problem

$$-\frac{\mathrm{d}^2 p(s)}{\mathrm{d}s^2} + p(s) = x(s), \quad s \in (0, 1),$$
$$p(s) = 0, \quad s \in \{0, 1\}. \tag{11}$$

It consists of recovering the unknown function $x \in L^\infty(D)$ from discrete, noisefree observations $y = Ax \in \mathbb{R}^K$, where $A = \mathcal{O} \circ G^{-1}$. Here, $G = -\frac{\mathrm{d}^2}{\mathrm{d}^2 s} + \mathrm{Id}$ denotes the differential operator on $\mathcal{D}(G) = H_0^1([0, 1])$ and $\mathcal{O} : H_0^1(D) \to \mathbb{R}^K$ denotes the discrete observation operator evaluating a function $p \in H_0^1([0, 1])$ at $K = 64$ equidistant observation points $s_k = k/K$, $k = 1, \ldots, K$, i.e.,

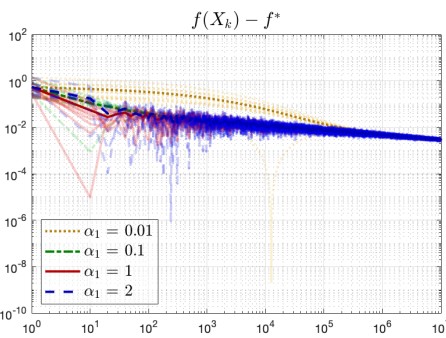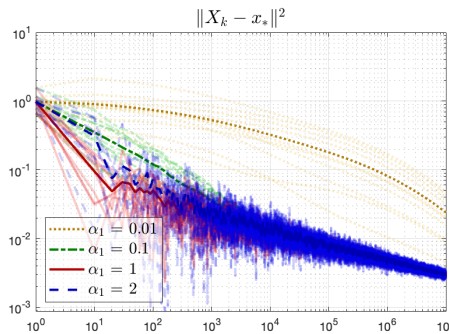

Figure 11: Left: pathwise optimality gap $f(X_k) - f(x_*)$. Right: pathwise squared error to the minimum-norm solution $\|X_k - x_*\|^2$. Each curve represents one of 10 independent runs of reg-SGD, each of length $N = 10^7$. The shaded lines depict individual runs of reg-SGD. The solid lines show the average errors for reg-SGD. The different colors correspond to various choices of $\alpha_1 \in \{0.01, 0.1, 1, 2\}$.

$\mathcal{O}p(\cdot) = (p(s_1), \ldots, p(s_K))^\top$. The ground truth right-hand side $x^\dagger$ used to generate the data is simulated as a random function

$$x^\dagger(s) = \sum_{i=1}^{100} \frac{\sqrt{2}}{\pi} \xi_i \sin(i\pi s), \quad \xi_i \sim \mathcal{N}(0, i^{-4}).$$

*Implementation details:* We numerically approximate the solution operator $G^{-1}$ on a grid $D_\delta \subset [0,1]$ with mesh size $\delta = 2^{-8}$ and represent the unknown function as a vector $x^\dagger \in \mathbb{R}^d$ with $d = 2^8$. The resulting discretized forward model is then given by a matrix $A \in \mathbb{R}^{K \times d}$ and the inverse problem reduces to solving the least-squares problem:

$$\min_{x \in \mathbb{R}^d} f(x), \quad f(x) := \frac{1}{2}\|Ax - y\|^2,$$

where $y = (p(s_1), \ldots, p(s_K)) \in \mathbb{R}^K$ contains the discrete measurements associated with (11).

We implemented both SGD and reg-SGD by partitioning the forward operator $A \in \mathbb{R}^{K \times d}$ into rows $A_i \in \mathbb{R}^{1 \times d}$, $i = 1, \ldots, K$. Hence, $A_i x$ corresponds to the discretized ODE solution at location $s_i$. At each iteration, a batch of 16 locations $(s_{i_1}, \ldots, s_{i_{16}})$ are sampled uniformly at random, and the gradient of $f$ is approximated by $\nabla f(x) \approx \frac{1}{16} \sum_{j=1}^{16} A_{i_j}^\top (A_{i_j} x - y_{i_j}) \in \mathbb{R}^d$ and additionally perturbed by independent noise following a multivariate normal distribution with zero mean and

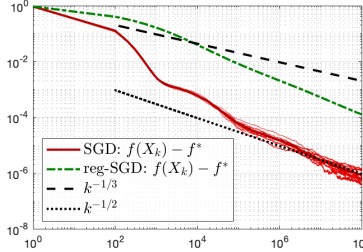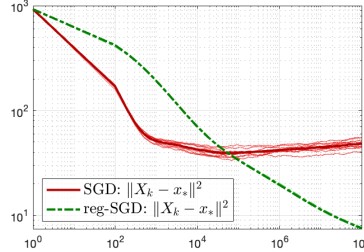

Figure 12: Left: pathwise optimality gap $f(X_k) - f(x_*)$. Right: pathwise squared error to the minimum-norm solution $\|X_k - x_*\|^2$. Each curve represents one of 10 independent runs, each of length $N = 10^7$. The red shaded lines depict individual runs of SGD, while the green dash-dotted shaded lines correspond to reg-SGD. The red solid line shows the average error across runs for SGD, the green bold dash-dotted line shows the average for reg-SGD, and the black dashed line indicates the theoretical convergence rate.

covariance $0.001^2 \cdot \mathrm{Id}$. For SGD we chose the step-size schedule $\alpha_k = 100k^{-1/2}$. For reg-SGD we chose $\alpha_k = 100k^{-2/3}$ and regularization $\lambda_k = 0.001k^{-1/3}$. We initialize both algorithms for each repetition at zero.

In Figure 12, we compare the expected and pathwise optimality gap (left) as well as the $L^2$ and pathwise error to the minimum-norm solution (right). While SGD shows fast convergence in terms of the optimality gap, it again fails to converge to the minimum-norm solution. In contrast, reg-SGD slows down the convergence in terms of the optimality gap, but safely reconstructs the minimum-norm solution. In Figure 13 (left), we plot the reconstruction of the unknown right-hand side $x^\dagger$ resulting from the minimum-norm solution $x_* = A^\dagger y$, and from the last iterates of SGD and reg-SGD. Moreover, in Figure 13 (right) we plot the corresponding ODE solutions when solving (11) with the estimated right-hand side.

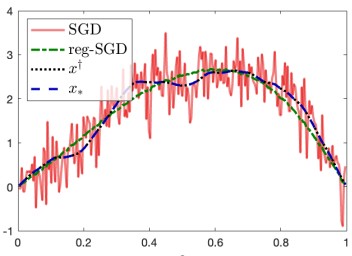 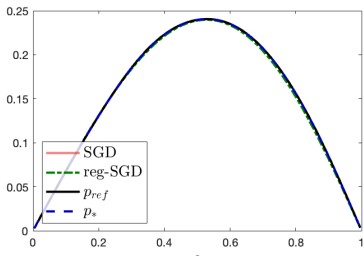

Figure 13: Left: reconstruction of $x^\dagger$ using the minimum-norm solution $x_* = A^\dagger y$, where $A^\dagger$ is the Moore-Penrose inverse of A, the last iterate of reg-SGD and of SGD. Right: corresponding ODE solutions of (11).

## B  Auxiliary results

In the following section, we provide a list of auxiliary results which are needed in the proofs of our main results.

**Lemma B.1.** *Suppose that $f$ satisfies Assumption 1.1, then the following statements hold true:*

*(i) For all $\lambda, \lambda' \geq 0$ it holds that*

$$f_\lambda(x_\lambda) \leq f_{\lambda'}(x_{\lambda'}) + \frac{\lambda - \lambda'}{2}\|x_{\lambda'}\|_{\mathcal{X}}^2 \,.$$

*(ii) For all $\lambda \geq \lambda' \geq 0$ it holds that*

$$0 \leq f_\lambda(x_\lambda) - f_{\lambda'}(x_{\lambda'}) \leq \frac{\lambda - \lambda'}{2}\|x_*\|_{\mathcal{X}}^2 \,.$$

*(iii) For all $\lambda \geq 0$ it holds that*

$$f(x) - f(x_*) \leq f_\lambda(x) - f_\lambda(x_\lambda) + \frac{\lambda}{2}\|x_*\|_{\mathcal{X}}^2.$$

*Proof.* The first assertion is a direct consequence of $f_\lambda(x_\lambda) \leq f_\lambda(x_{\lambda'}) = f(x_{\lambda'}) + \frac{\lambda}{2}\|x_{\lambda'}\|_{\mathcal{X}}^2$, since $x_\lambda$ is the minimum of $f_\lambda$. The second assertion follows from $f_\lambda(x) \geq f_{\lambda'}(x)$ for all $x \in \mathcal{X}$ and $\|x_{\lambda'}\|_{\mathcal{X}} \leq \|x_*\|_{\mathcal{X}}$. For the third assertion we use (ii) with $\lambda' \to 0$ together with $f(x) \leq f_\lambda(x)$ for all $x \in \mathcal{X}$. □

**Lemma B.2.** *Let $f$ be $L$-smooth, then $f_\lambda$ is $L + \lambda$-smooth for any $\lambda \geq 0$.*

*Proof.* For arbitrary $x, y \in \mathcal{X}$ we apply triangle inequality to deduce

$$\|\nabla f_\lambda(x) - \nabla f_\lambda(y)\|_{\mathcal{X}} = \|\nabla f(x) - \nabla f(y) + \lambda(x - y)\|_{\mathcal{X}} \leq L\|x - y\|_{\mathcal{X}} + \lambda\|x - y\|_{\mathcal{X}} \,.$$

□

Note that by $L$-smoothness the descent condition holds, meaning that for any $x, y \in \mathcal{X}$ we have

$$f(y) \leq f(x) + \langle \nabla f(x), y - x \rangle_{\mathcal{X}} + \frac{L}{2} \|x - y\|_{\mathcal{X}}^2. \tag{12}$$

The following lemma is similar to [7, Lemma 3]. For completeness we give a proof.

**Lemma B.3.** *Under Assumption 1.1 the following estimates are satisfied for all $x \in \mathcal{X}$ and $\lambda \geq 0$:*

(i) $f(x) - f(x_*) \leq f_\lambda(x) - f_\lambda(x_\lambda) + \frac{\lambda}{2} \|x_*\|_{\mathcal{X}}^2$

(ii) $\|x - x_\lambda\|_{\mathcal{X}}^2 \leq \frac{2(f_\lambda(x) - f_\lambda(x_\lambda))}{\lambda}$

*Proof.* (i): For arbitrary $x \in \mathcal{X}$ we have

$$
\begin{aligned}
f(x) - f(x_*) &= f_\lambda(x) - f_\lambda(x_*) + \frac{\lambda}{2}(\|x_*\|_{\mathcal{X}}^2 - \|x\|_{\mathcal{X}}^2) \\
&= f_\lambda(x) - f_\lambda(x_\lambda) + f_\lambda(x_\lambda) - f_\lambda(x_*) + \frac{\lambda}{2}(\|x_*\|_{\mathcal{X}}^2 - \|x\|_{\mathcal{X}}^2) \\
&\leq f_\lambda(x) - f_\lambda(x_\lambda) + \frac{\lambda}{2} \|x_*\|_{\mathcal{X}}^2.
\end{aligned}
$$

(ii): The second assertion follows from the $\lambda$-strong convexity of $f_\lambda$ and $\nabla f_\lambda(x_\lambda) = 0$. $\square$

**Lemma B.4.** *Let $p > 0$ and $\lambda_k = \frac{1}{k^p}$, $k \in \mathbb{N}$. Then for all $k \in \mathbb{N}$ one has*

$$\frac{p}{(k+1)^{p+1}} \leq \lambda_k - \lambda_{k+1} \leq \frac{p}{k^{p+1}}$$

*and*

$$\frac{\lambda_{k-1}}{\lambda_k} = 1 + \frac{p}{k} + o\left(\frac{1}{k}\right).$$

*Proof.* We define $\varphi(s) = s^{-p}$, $s \in (0, \infty)$, and note that $\varphi'(s) = -ps^{-(p+1)}$. By the mean value theorem, for all $k \in \mathbb{N}$ there exists a $c \in [k, k+1]$ such that

$$\lambda_k - \lambda_{k+1} = \varphi(k) - \varphi(k+1) = -\varphi'(c)(k + 1 - k) = \frac{p}{c^{p+1}}.$$

The first assertion follows by the monotonicity of $s \mapsto 1/s^{p+1}$. For the second assertion, we use Taylor's approximation theorem at $s = 1$ to get

$$
\begin{aligned}
\frac{\lambda_{k-1}}{\lambda_k} = \left(\frac{k-1}{k}\right)^{-p} = \varphi\left(\frac{k-1}{k}\right) &= \varphi(1) + \varphi'(1)\left(\frac{k-1}{k} - 1\right) + o\left(\left|\frac{k-1}{k} - 1\right|\right) \\
&= 1 + \frac{p}{k} + o\left(\frac{1}{k}\right).
\end{aligned}
$$

$\square$

**Lemma B.5** (Robbins-Siegmund theorem, see Theorem 1 in [62]). *Let $(\mathcal{F}_k)_{k \in \mathbb{N}}$ be a filtration and $(X_k)_{k \in \mathbb{N}}, (Y_k)_{k \in \mathbb{N}}$, and $(Z_k)_{k \in \mathbb{N}}$ be $(\mathcal{F}_k)_{k \in \mathbb{N}}$-adapted sequences of non-negative random variables. Let $(\gamma_k)_{k \in \mathbb{N}}$ be a sequence of non-negative reals and assume that*

(i) $\prod_{k=1}^{\infty}(1 + \gamma_k) < \infty$,

(ii) $\sum_{k=1}^{\infty} Z_k < \infty$, *almost surely, and*

(iii) $\mathbb{E}[Y_{k+1} \mid \mathcal{F}_k] \leq (1 + \gamma_k)Y_k - X_k + Z_k$, *almost surely for all $k \in \mathbb{N}$.*

*Then $\sum_{k=1}^{\infty} X_k < \infty$ and $(Y_k)_{k \in \mathbb{N}}$ converges almost surely.*

We will use the following two versions of the Robbins-Siegmund theorem.

**Corollary B.6.** *Let $(\mathcal{F}_k)_{k\in\mathbb{N}}$ be a filtration, let $(z_k)_{k\in\mathbb{N}}$ be a summable sequence of non-negative reals and let $(Y_k)_{k\in\mathbb{N}}$ be an $(\mathcal{F}_k)_{k\in\mathbb{N}}$-adapted sequence that is uniformly bounded from below. Assume that for all $k \in \mathbb{N}$*

$$\mathbb{E}[Y_{k+1} \mid \mathcal{F}_k] \leq Y_k + z_k. \tag{13}$$

*Then $(Y_k)_{k\in\mathbb{N}}$ converges almost surely.*

*Proof.* Let $C \geq 0$ be a constant such that for all $k \in \mathbb{N}$ one has $Y_k \geq -C$, almost surely. Set $(\tilde{Y}_k)_{k\in\mathbb{N}} = (Y_k + C)_{k\in\mathbb{N}}$ and note that (13) still holds when replacing $(Y_k)_{k\in\mathbb{N}}$ by $(\tilde{Y}_k)_{k\in\mathbb{N}}$. Thus, the statement follows from Lemma B.5 for the choice $\gamma_k \equiv 0$, $X_k \equiv 0$ and $(Z_k)_{k\in\mathbb{N}} = (z_k)_{k\in\mathbb{N}}$. □

**Corollary B.7.** *Let $(\mathcal{F}_k)_{k\in\mathbb{N}}$ be a filtration and $(Y_k)_{k\in\mathbb{N}}$, $(A_k)_{k\in\mathbb{N}}$, $(B_k)_{k\in\mathbb{N}}$ and $(C_k)_{k\in\mathbb{N}}$ be non-negative and adapted processes satisfying almost surely that*

$$\sum_{k=1}^{\infty} A_k = \infty \quad , \quad \sum_{k=1}^{\infty} B_k < \infty \quad and \quad \sum_{k=1}^{\infty} C_k < \infty.$$

*Moreover, suppose that for all $k \in \mathbb{N}$ one has almost surely that*

$$\mathbb{E}[Y_{k+1} \mid \mathcal{F}_k] \leq (1 + C_k - A_k)Y_k + B_k.$$

*Then $Y_k \to 0$ holds almost surely as $k \to \infty$.*

*Proof.* The proof follows the same lines as the proof of Lemma A.2 in [72]. For completeness, we provide the full details. Compared to Lemma B.5, we have $Y_k = Y_k$, $X_k = A_k Y_k$, $Z_k = B_k$ and $\gamma_k = C_k$. Using Lemma B.5 we obtain the existence of $Y_\infty$ which is almost surely finite, integrable and satisfies $Y_n \to Y_\infty$ almost surely. Additionally, we have that $\sum_{k=1}^{\infty} X_k = \sum A_k Y_k < \infty$ implying that $\liminf_{k\to\infty} Y_k = 0$, where we have used the assumption $\sum_{k=1}^{\infty} A_k = \infty$ almost surely. Since the limit inferior and limit coincide for converging sequences, the assertion follows by

$$Y_\infty = \lim_{k\to\infty} Y_k = \liminf_{k\to\infty} Y_k = 0 \quad \text{almost surely} .$$

□

# C  Finite-sum problems

In this section, we prove (6) from Example 1.3 in the introduction. We consider the finite-sum optimization problem

$$\min_{x\in\mathbb{R}^d} f(x) = \frac{1}{N} \sum_{i=1}^{N} f_i(x),$$

where, for all $i = 1, \ldots, N$, $f_i : \mathbb{R}^d \to \mathbb{R}$ is convex and $L_i$-smooth. The mini-batch estimator with mini-batch size $M \in \mathbb{N}$ is defined via $g_k = \frac{1}{M} \sum_{i\in M} \nabla f_{I_{i,k}}(X_{k-1})$, for all $k \in \mathbb{N}$, where $(I_{i,k})_{i,k\in\mathbb{N}}$ is a family of iid. random variables that are uniformly distributed on $\{1, \ldots, N\}$. The corresponding gradient noise is defined as $D_k = \frac{1}{M} \sum_{i\in M} (\nabla f_{I_{i,k}}(X_{k-1}) - \nabla f(X_{k-1}))$. We show that in the finite-sum situation the ABC-condition, Assumption 1.2, is satisfied. The following lemma is a version of [34, Lemma 4.20] with improved constants.

**Lemma C.1.** *The sequence $(D_k)_{k\in\mathbb{N}}$ satisfies for all $k \in \mathbb{N}$*

$$\mathbb{E}\left[\|D_k\|^2 \mid \mathcal{F}_{k-1}\right] \leq \frac{4\bar{L}}{M}\big(f(X_{k-1}) - f(x_*)\big) + \frac{2\sigma_*^2}{M},$$

*where $\bar{L} = \frac{1}{N} \sum_{i=1}^{N} L_i$ and $\sigma_*^2 = \frac{1}{N} \sum_{i=1}^{N} \|\nabla f_i(x_*)\|^2$.*

*Proof.* Since, for all $i \in \{1, \ldots, N\}$, $f_i$ is convex and $L_i$-smooth we get for all $x, y \in \mathbb{R}^d$ that

$$f_i(x) - f_i(y) \leq \langle \nabla f_i(y), x - y \rangle + \frac{L_i}{2}\|x - y\|^2.$$

For fixed $y \in \mathbb{R}^d$ let

$$\varphi_i(x) = f_i(x) - f_i(y) - \langle \nabla f_i(y), x - y \rangle.$$

Due to convexity of $f_i$, $\varphi_i$ is non-negative. Moreover, $\nabla \varphi_i(x) = \nabla f_i(x) - \nabla f_i(y)$ is $L_i$-Lipschitz. Thus, for $z = x - \frac{\nabla \varphi_i(x)}{L_i}$

$$
\begin{aligned}
0 \le \varphi_i(z) &= \varphi_i(x) - \langle \nabla \varphi_i(x), \frac{\nabla \varphi_i(x)}{L_i} \rangle + \frac{L_i}{2} \| \frac{\nabla \varphi_i(x)}{L_i} \|^2 \\
&= f_i(x) - f_i(y) - \langle \nabla f_i(y), x - y \rangle - \frac{1}{2L_i} \| \nabla f_i(x) - \nabla f_i(y) \|^2,
\end{aligned}
$$

which yields

$$\| \nabla f_i(x) - \nabla f_i(y) \|^2 \le 2L_i (f_i(x) - f_i(y) - \langle \nabla f_i(y), x - y \rangle). \tag{14}$$

Thus,

$$
\begin{aligned}
\mathbb{E}\left[ \|D_k\|^2 \mid \mathcal{F}_{k-1} \right] &= \frac{1}{NM} \sum_{i=1}^{N} \| \nabla f_i(X_{k-1}) - \nabla f(X_{k-1}) \|^2 \\
&= \frac{1}{NM} \sum_{i=1}^{N} \| \nabla f_i(X_{k-1}) - \nabla f_i(x_*) - \nabla f(X_{k-1}) + \nabla f_i(x_*) \|^2 \\
&= \frac{2}{M} \sigma_*^2 + \frac{2}{NM} \sum_{i=1}^{N} \| \nabla f_i(X_{k-1}) - \nabla f_i(x_*) - \nabla f(X_{k-1}) \|^2
\end{aligned}
$$

Since $\frac{1}{N} \sum_{i=1}^{N} \nabla f_i(X_{k-1}) - \nabla f_i(x_*) = \nabla f(X_{k-1})$, we can use (14) with $x = X_{k-1}$ and $y = x_*$ to get

$$
\begin{aligned}
\frac{1}{N} \sum_{i=1}^{N} \| \nabla f_i(X_{k-1}) - \nabla f_i(x_*) - \nabla f(X_{k-1}) \|^2 &\le \frac{1}{N} \sum_{i=1}^{N} \| \nabla f_i(X_{k-1}) - \nabla f_i(x_*) \|^2 \\
&\le 2\bar{L} f(X_{k-1}) - f(x_*) - \underbrace{\frac{1}{N} \sum_{i=1}^{N} \langle \nabla f_i(x_*), X_{k-1} - x_* \rangle}_{\langle \nabla f(x_*), X_{k-1} - x_* \rangle = 0}.
\end{aligned}
$$

$\square$

# D  Proofs of the main results

As a first step, we derive an iterative bound for the optimality gap of the regularized objective function

$$E_k := f_{\lambda_{k+1}}(X_k) - f_{\lambda_{k+1}}(x_{\lambda_{k+1}}), \quad k \in \mathbb{N}_0. \tag{15}$$

Given Lemma B.3, this process $(E_k)_{k \in \mathbb{N}_0}$ serves as Lyapunov function for computing the convergence rates stated in Section 2.

**Proposition D.1.** *Suppose that Assumption 1.1 and Assumption 1.2 are fulfilled and and let $(X_k)_{k \in \mathbb{N}_0}$ be generated by (4) with predictable (random) step-sizes and regularization parameters that are uniformly bounded from above and such that $(\lambda_k)_{k \in \mathbb{N}}$ is almost surely decreasing. For $k \in \mathbb{N}$ denote by $\mathbb{A}_k = \{ \alpha_k \le \frac{2}{L + \lambda_k} \} \in \mathcal{F}_{k-1}$. Then, for all $k \in \mathbb{N}$,*

$$
\begin{aligned}
\mathbb{E}[\mathbb{1}_{\mathbb{A}_k} E_k \mid \mathcal{F}_{k-1}] &\le \left( 1 - 2\lambda_k \alpha_k \left( 1 - \frac{L + \lambda_k}{2} \alpha_k \right) + \frac{L + \lambda_k}{2} \alpha_k^2 A \right) \mathbb{1}_{\mathbb{A}_k} E_{k-1} \\
&\quad + \frac{\lambda_k - \lambda_{k+1}}{2} \| x_* \|_{\mathcal{X}}^2 + \frac{L + \lambda_k}{2} \alpha_k^2 \left( A \frac{\lambda_k}{2} \| x_* \|_{\mathcal{X}}^2 + C \right).
\end{aligned}
$$

*Proof.* Using Assumption 1.2, the property $f \leq f_{\lambda_k}$, and the descent condition (12) applied to the $(L + \lambda_k)$-smooth function $f_{\lambda_k}$, yields, for $k \in \mathbb{N}$,

$$\mathbb{E}[\mathbb{1}_{\mathbb{A}_k} f_{\lambda_k}(X_k) \mid \mathcal{F}_{k-1}] \leq \mathbb{1}_{\mathbb{A}_k} \Big( f_{\lambda_k}(X_{k-1}) - \alpha_k \Big(1 - \frac{L + \lambda_k}{2}\alpha_k\Big) \|\nabla f_{\lambda_k}(X_{k-1})\|_{\mathcal{X}}^2$$
$$+ \frac{L + \lambda_k}{2}\alpha_k^2 \mathbb{E}[\|D_k\|_{\mathcal{X}}^2 \mid \mathcal{F}_{k-1}] \Big)$$
$$\leq \mathbb{1}_{\mathbb{A}_k} \Big( f_{\lambda_k}(X_{k-1}) - \alpha_k \Big(1 - \frac{L + \lambda_k}{2}\alpha_k\Big) \|\nabla f_{\lambda_k}(X_{k-1})\|_{\mathcal{X}}^2$$
$$+ \frac{L + \lambda_k}{2}\alpha_k^2 \big(A(f(X_{k-1}) - f(x_*)) + C\big) \Big)$$
$$\leq \mathbb{1}_{\mathbb{A}_k} \Big( f_{\lambda_k}(X_{k-1}) - \alpha_k \Big(1 - \frac{L + \lambda_k}{2}\alpha_k\Big) \|\nabla f_{\lambda_k}(X_{k-1})\|_{\mathcal{X}}^2$$
$$+ \frac{L + \lambda_k}{2}\alpha_k^2 \big(A(f_{\lambda_k}(X_{k-1}) - f_{\lambda_k}(x_{\lambda_k})) + A\frac{\lambda_k}{2}\|x_*\|_{\mathcal{X}}^2 + C\big) \Big),$$

where in the last step we also used Lemma B.1 (iii). Since each $f_{\lambda_k}$ is $\lambda_k$-strongly convex, it satisfies the Polyak-Łojasiewicz inequality

$$f_{\lambda_k}(x) - f_{\lambda_k}(x_{\lambda_k}) \leq \frac{1}{2\lambda_k}\|\nabla f_{\lambda_k}(x)\|_{\mathcal{X}}^2, \quad x \in \mathcal{X}. \tag{16}$$

Thus,

$$\mathbb{E}[\mathbb{1}_{\mathbb{A}_k} f_{\lambda_k}(X_k) \mid \mathcal{F}_{k-1}] \leq \mathbb{1}_{\mathbb{A}_k} \Big( f_{\lambda_k}(X_{k-1}) - 2\alpha_k \lambda_k \Big(1 - \frac{L + \lambda_k}{2}\alpha_k\Big)(f_{\lambda_k}(X_{k-1}) - f_{\lambda_k}(x_{\lambda_k}))$$
$$+ \frac{L + \lambda_k}{2}\alpha_k^2 \big(A(f_{\lambda_k}(X_{k-1}) - f_{\lambda_k}(x_{\lambda_k})) + A\frac{\lambda_k}{2}\|x_*\|_{\mathcal{X}}^2 + C\big) \Big).$$

Next, we observe that

$$f_{\lambda_{k+1}}(X_k) - f_{\lambda_{k+1}}(x_{\lambda_{k+1}}) = f_{\lambda_k}(X_k) - f_{\lambda_k}(x_{\lambda_k}) + f_{\lambda_{k+1}}(X_k) - f_{\lambda_k}(X_k)$$
$$+ f_{\lambda_k}(x_{\lambda_k}) - f_{\lambda_{k+1}}(x_{\lambda_{k+1}})$$
$$\leq f_{\lambda_k}(X_k) - f_{\lambda_k}(x_{\lambda_k}) + f_{\lambda_k}(x_{\lambda_k}) - f_{\lambda_{k+1}}(x_{\lambda_{k+1}}),$$

since $f_{\lambda_{k+1}}(X_k) - f_{\lambda_k}(X_k) \leq 0$. Combining the previous computations and using Lemma B.1 (ii) yields

$$\mathbb{E}[\mathbb{1}_{\mathbb{A}_k} E_k \mid \mathcal{F}_{k-1}] \leq \Big(1 - 2\lambda_k \alpha_k \Big(1 - \frac{L + \lambda_k}{2}\alpha_k\Big)\Big)\mathbb{1}_{\mathbb{A}_k} E_{k-1} + f_{\lambda_k}(x_{\lambda_k}) - f_{\lambda_{k+1}}(x_{\lambda_{k+1}})$$
$$+ \frac{L + \lambda_k}{2}\alpha_k^2 \big(\mathbb{1}_{\mathbb{A}_k} A(f_{\lambda_k}(X_{k-1}) - f_{\lambda_k}(x_{\lambda_k})) + A\frac{\lambda_k}{2}\|x_*\|_{\mathcal{X}}^2 + C\big)$$
$$\leq \Big(1 - 2\lambda_k \alpha_k \Big(1 - \frac{L + \lambda_k}{2}\alpha_k\Big)\Big)\mathbb{1}_{\mathbb{A}_k} E_{k-1} + \frac{\lambda_k - \lambda_{k+1}}{2}\|x_*\|_{\mathcal{X}}^2$$
$$+ \frac{L + \lambda_k}{2}\alpha_k^2 \big(\mathbb{1}_{\mathbb{A}_k} A E_{k-1} + A\frac{\lambda_k}{2}\|x_*\|_{\mathcal{X}}^2 + C\big).$$

$\qquad\qquad\square$

With the help of the energy function $(E_k)_{k \in \mathbb{N}_0}$, we can bound the optimality gap of the true objective function, as well as the distance to the unique minimizer of the regularized objective function. For this, we rephrase Lemma B.3 in the notation used in this section.

**Lemma D.2.** *Suppose that Assumption 1.1 is fulfilled and let $(X_k)_{k \in \mathbb{N}_0}$ be generated by (4). Then the following estimates are satisfied for all $k \in \mathbb{N}$:*

*(i)* $f(X_k) - f(x_*) \leq E_k + \frac{\lambda_{k+1}}{2}\|x_*\|_{\mathcal{X}}^2,$

*(ii)* $\|X_k - x_{\lambda_{k+1}}\|_{\mathcal{X}}^2 \leq \frac{2E_k}{\lambda_{k+1}}.$

## D.1 General convergence result

First, we prove the general convergence results. We emphasize that no boundedness assumption is imposed on the reg-SGD scheme. In fact, in the proof of Theorem 2.1 we will show that Assumptions 1.1-1.2 together with (17) imply that $\sup_{k\in\mathbb{N}_0} \|X_k\|_{\mathcal{X}}^2 < \infty$ almost surely. The proof uses ideas from Theorem 4.1 in [56] and Lemma 3.1 in [28]. Due to the discretization error, we introduce and analyze a combined Lyapunov function $(\varphi_k + E_k)_{k\in\mathbb{N}}$, where $\varphi_k = \|X_k - x_*\|_{\mathcal{X}}^2$ and $E_k$ is defined in (15), in order to prove a descent step.

### D.1.1 Proof of Theorem 2.1

Let us recall the statements. We note that using a stopping time argument one can lift the boundedness assumption on the step-sizes and regularization parameters. However, proving this generalization requires a lot of heavy notation and technical arguments.

**Theorem D.3** (Almost sure convergence). *Suppose that Assumption 1.1 and Assumption 1.2 are fulfilled and let $(X_k)_{k\in\mathbb{N}_0}$ be generated by (4) with predictable (random) step-sizes and regularization parameters that are uniformly bounded from above. Moreover, we assume that almost surely $(\lambda_k)_{k\in\mathbb{N}}$ is decreasing to 0 and*

$$\sum_{k\in\mathbb{N}} \alpha_k \lambda_k = \infty \quad , \quad \sum_{k\in\mathbb{N}} \alpha_k^2 < \infty \quad and \quad \sum_{k\in\mathbb{N}} \alpha_k \lambda_k \big(\|x_*\|_{\mathcal{X}}^2 - \|x_{\lambda_k}\|_{\mathcal{X}}^2\big) < \infty. \tag{17}$$

*Then $\lim_{k\to\infty} X_k = x_*$ almost surely.*

*Proof.* For $k\in\mathbb{N}_0$ let $\varphi_k = \|X_k - x_*\|_{\mathcal{X}}^2$. Then, for all $k\in\mathbb{N}$,

$$\mathbb{E}[\varphi_k \mid \mathcal{F}_{k-1}] \leq \varphi_{k-1} - 2\alpha_k \langle \nabla f_{\lambda_k}(X_{k-1}), X_{k-1} - x_*\rangle_{\mathcal{X}} + \alpha_k^2 \|\nabla f_{\lambda_k}(X_{k-1})\|_{\mathcal{X}}^2$$
$$+ \alpha_k^2 \Big(A(f_{\lambda_k}(X_{k-1}) - f_{\lambda_k}(x_{\lambda_k})) + A\frac{\lambda_k}{2}\|x_*\|_{\mathcal{X}}^2 + C\Big), \tag{18}$$

where we used Assumption 1.2 and Lemma B.1 (iii). Strong convexity of $f_{\lambda_k}$ yields

$$f_{\lambda_k}(x_*) \geq f_{\lambda_k}(X_{k-1}) + \langle \nabla f_{\lambda_k}(X_{k-1}), x_* - X_{k-1}\rangle_{\mathcal{X}} + \frac{\lambda_k}{2}\|X_{k-1} - x_*\|_{\mathcal{X}}^2$$

$$\geq f_{\lambda_k}(x_{\lambda_k})) + \langle \nabla f_{\lambda_k}(X_{k-1}), x_* - X_{k-1}\rangle_{\mathcal{X}} + \frac{\lambda_k}{2}\|X_{k-1} - x_*\|_{\mathcal{X}}^2.$$

Since, $f(x_*) \leq f(x_{\lambda_k})$ this implies

$$\frac{\lambda_k}{2}\|x_*\|_{\mathcal{X}}^2 \geq \frac{\lambda_k}{2}\|x_{\lambda_k}\|_{\mathcal{X}}^2 + \langle \nabla f_{\lambda_k}(X_{k-1}), x_* - X_{k-1}\rangle_{\mathcal{X}} + \frac{\lambda_k}{2}\|X_{k-1} - x_*\|_{\mathcal{X}}^2,$$

so that

$$\langle \nabla f_{\lambda_k}(X_{k-1}), X_{k-1} - x_*\rangle_{\mathcal{X}} \geq \frac{\lambda_k}{2}(\|x_{\lambda_k}\|_{\mathcal{X}}^2 - \|x_*\|_{\mathcal{X}}^2) + \frac{\lambda_k}{2}\|X_{k-1} - x_*\|_{\mathcal{X}}^2. \tag{19}$$

Combining (18) and (19) gives

$$\mathbb{E}[\varphi_k \mid \mathcal{F}_{k-1}] \leq (1 - \alpha_k \lambda_k)\varphi_{k-1} + \alpha_k \lambda_k(\|x_*\|_{\mathcal{X}}^2 - \|x_{\lambda_k}\|_{\mathcal{X}}^2) + \alpha_k^2 \|\nabla f_{\lambda_k}(X_{k-1})\|_{\mathcal{X}}^2$$
$$+ \alpha_k^2\big(A(f_{\lambda_k}(X_{k-1}) - f_{\lambda_k}(x_{\lambda_k})) + A\frac{\lambda_k}{2}\|x_*\|_{\mathcal{X}}^2 + C\big)$$
$$\leq (1 - \alpha_k \lambda_k)\varphi_{k-1} + \alpha_k \lambda_k(\|x_*\|_{\mathcal{X}}^2 - \|x_{\lambda_k}\|_{\mathcal{X}}^2)$$
$$+ \alpha_k^2\big((A + 2L + 2\lambda_k)(f_{\lambda_k}(X_{k-1}) - f_{\lambda_k}(x_{\lambda_k})) + A\frac{\lambda_k}{2}\|x_*\|_{\mathcal{X}}^2 + C\big),$$

where in the last step we used that analogously to (5) one has

$$\|\nabla f_{\lambda_k}(x)\|_{\mathcal{X}}^2 \leq 2(L + \lambda_k)(f_{\lambda_k}(x) - f(x_{\lambda_k})) \quad \text{for all } x \in \mathcal{X}.$$

Now, recall that Proposition D.1 gives that for all $k\in\mathbb{N}$

$$\mathbb{E}[\mathbb{1}_{\mathbb{A}_k} E_k \mid \mathcal{F}_{k-1}] \leq \Big(1 - 2\lambda_k\alpha_k\Big(1 - \frac{L+\lambda_k}{2}\alpha_k\Big) + \frac{L+\lambda_k}{2}\alpha_k^2 A\Big)\mathbb{1}_{\mathbb{A}_k} E_{k-1}$$
$$+ \frac{\lambda_k - \lambda_{k+1}}{2}\|x_*\|_{\mathcal{X}}^2 + \frac{L+\lambda_k}{2}\alpha_k^2\Big(A\frac{\lambda_k}{2}\|x_*\|_{\mathcal{X}}^2 + C\Big),$$

where $E_k = f_{\lambda_{k+1}}(X_k) - f_{\lambda_{k+1}}(x_{\lambda_{k+1}})$. Fix $N \in \mathbb{N}$ and for $k \geq N$ denote $\mathbb{B}_k(N) = \{\alpha_i \leq \frac{1}{L+\lambda_i} : i = N, \ldots, k\}$. Then, for all $k > N$

$$
\begin{aligned}
\mathbb{E}[\mathbf{1}_{\mathbb{B}_k(N)}(\varphi_k + E_k) \mid \mathcal{F}_{k-1}] &\leq \tau_k \mathbf{1}_{\mathbb{B}_{k-1}(N)}(\varphi_{k-1} + E_{k-1}) + \alpha_k \lambda_k (\|x_*\|_{\mathcal{X}}^2 - \|x_{\lambda_k}\|_{\mathcal{X}}^2) \\
&\quad + \frac{\lambda_k - \lambda_{k+1}}{2}\|x_*\|_{\mathcal{X}}^2 + \alpha_k^2 \Big(A\frac{\lambda_k}{2}\|x_*\|_{\mathcal{X}}^2 + C\Big)\Big(1 + \frac{L+\lambda_k}{2}\Big),
\end{aligned} \tag{20}
$$

where

$$
\tau_k = \max\Big(1 - \alpha_k\lambda_k, 1 - 2\lambda_k\alpha_k\Big(1 - \frac{L+\lambda_k}{2}\alpha_k\Big) + \Big(\frac{L+\lambda_k}{2}A + A + 2L + 2\lambda_k\Big)\alpha_k^2\Big)
$$

and we have used that $\varphi_{k-1} + E_{k-1} \geq 0$ and $\mathbb{B}_{k-1}(N) \supset \mathbb{B}_k(N)$. On the event $\mathbb{B}_{k-1}(N)$ we have

$$
\tau_k \leq 1 - \underbrace{\alpha_k\lambda_k}_{=:A_k} + \underbrace{\Big(\frac{L+\lambda_k}{2}A + A + 2L + 2\lambda_k\Big)\alpha_k^2}_{=:C_k}, \tag{21}
$$

where by assumption $\sum_{k\in\mathbb{N}} C_k < \infty$ and $\sum_{k\in\mathbb{N}} A_k = \infty$ almost surely. Now, we can apply Corollary B.7 for the process $(\mathbf{1}_{\mathbb{B}_k(N)}(\varphi_k + E_k))_{k\geq N}$ to deduce that, on $\mathbb{B}_\infty(N) = \bigcap_{k\geq N}\mathbb{B}_k(N)$, one has $\varphi_k \to 0$ almost surely as $k \to \infty$. Since $\alpha_k \to 0$ almost surely one has

$$
\mathbb{P}\Big(\bigcup_{N\in\mathbb{N}} \mathbb{B}_\infty(N)\Big) = 1
$$

and, thus, the proof of the theorem is finished. $\qquad\square$

### D.1.2  Proof of Theorem 2.2

We again reformulate the statement and provide the full proof of the general $L^2$-convergence.

**Theorem D.4** ($L^2$-convergence). *Suppose that Assumption 1.1 and Assumption 1.2 are fulfilled and let $(X_k)_{k\in\mathbb{N}_0}$ be generated by (4) with deterministic step-sizes and deterministic and decreasing regularization parameters $(\lambda_k)_{k\in\mathbb{N}}$. Moreover, assume that $\lambda_k \to 0$ and (17), or, alternatively,*

$$
\sum_{k\in\mathbb{N}} \alpha_k\lambda_k = \infty \quad, \quad \alpha_k = o(\lambda_k) \quad \text{and} \quad \lambda_k - \lambda_{k-1} = o(\alpha_k\lambda_k). \tag{22}
$$

*Then* $\lim_{k\to\infty} \mathbb{E}[\|X_k - x_*\|_{\mathcal{X}}^2] = 0$.

*Proof.* First, we prove the theorem assuming that (17) holds. By assumption, one has $\sum_{k\in\mathbb{N}} A_k = \infty$, $\sum_{k\in\mathbb{N}} C_k < \infty$, $\sum_{k\in\mathbb{N}} \alpha_k^2 < \infty$, $\sum_{k\in\mathbb{N}} \alpha_k\lambda_k(\|x_*\|_{\mathcal{X}}^2 - \|x_{\lambda_k}\|_{\mathcal{X}}^2) < \infty$ and $\sum_{k=1}^{\infty}(\lambda_k - \lambda_{k+1}) = \lambda_1 < \infty$, where $(A_k)$ and $(C_k)$ are defined in (21). Therefore, after taking expectations in (20), we can apply Corollary B.7 for the deterministic process $(Y_k)_{k\in\mathbb{N}} = (\mathbb{E}[\varphi_k + E_k])_{k\in\mathbb{N}}$ to deduce that $\mathbb{E}[\varphi_k] \to 0$ and $\mathbb{E}[E_k] \to 0$.

Let us now prove the statement under (22). Combining (20) with the fact that $\alpha_k \to 0$, there exist $C_1 > 0$ and $N \in \mathbb{N}$ such that for all $k \geq N$ one has

$$
\begin{aligned}
\mathbb{E}[\varphi_k + E_k \mid \mathcal{F}_{k-1}] &\leq (1 - C_1\alpha_k\lambda_k)(\varphi_{k-1} + E_{k-1}) + \alpha_k\lambda_k(\|x_*\|_{\mathcal{X}}^2 - \|x_{\lambda_k}\|_{\mathcal{X}}^2) \\
&\quad + \frac{\lambda_k - \lambda_{k+1}}{2}\|x_*\|_{\mathcal{X}}^2 + \alpha_k^2\Big(A\frac{\lambda_k}{2}\|x_*\|_{\mathcal{X}}^2 + C\Big)\Big(1 + \frac{L+\lambda_k}{2}\Big).
\end{aligned}
$$

Moreover, using that $\alpha_k^2 = o(\alpha_k\lambda_k)$, $\lambda_k - \lambda_{k+1} = o(\alpha_k\lambda_k)$ and $\|x_{\lambda_k}\|_{\mathcal{X}} \to \|x_*\|_{\mathcal{X}}$, for all $\varepsilon > 0$ there exists an $N \in \mathbb{N}$ such that for all $k \geq N$

$$
\mathbb{E}[\varphi_k + E_k \mid \mathcal{F}_{k-1}] \leq (1 - C_1\alpha_k\lambda_k)(\varphi_{k-1} + E_{k-1}) + \varepsilon\alpha_k\lambda_k. \tag{23}
$$

Rewriting (23) gives

$$
\mathbb{E}\Big[\varphi_k + E_k - \frac{\varepsilon}{C_1}\Big|\mathcal{F}_{k-1}\Big] \leq (1 - C_1\alpha_k\lambda_k)\Big(\varphi_{k-1} + E_{k-1} - \frac{\varepsilon}{C_1}\Big),
$$

so that, taking expectation and using $\sum_{k\in\mathbb{N}} \alpha_k\lambda_k = \infty$, we get

$$
\limsup_{k\to\infty} \mathbb{E}[\varphi_k + E_k] - \frac{\varepsilon}{C_1} \leq 0.
$$

The statement now follows from $\varepsilon \to 0$. $\qquad\square$

## D.2 Deterministic case: Convergence rate for reg-GD

Before discussing the convergence rates for reg-GD, we want to relate our analysis to the literature in convex optimization. For this purpose we formulate our task of finding the minimum-norm solution as constrained optimization problem in form of

$$\min_{x \in \mathcal{X}} \frac{1}{2} \|x\|_{\mathcal{X}}^2 \quad \text{s.t.} \quad x \in C := \arg\min_{y \in \mathcal{X}} f(y).$$

This naturally relates to the task of solving general variational inclusions of form

$$0 \in A(x) + N_C(x)$$

where $A$ denotes a (maximal) monotone operator and $N_C(x) = \{v \in \mathcal{X} : \langle v, w - x \rangle \leq 0 \quad \forall w \in C\}$ is the normal cone of a closed convex set $C$ at $x$. In our setting the operator $A(x) = \nabla_z \frac{1}{2} \|z\|_{\mathcal{X}}^2 \big|_{z=x} = x$ is strongly monotone. Another important class of problems studied in this context are hierarchical optimization problems of finding points in the set

$$S = \arg\min\{g(x) \mid x \in \arg\min f(x)\}$$

for two convex functions $g$ and $f$. This relates to our setting by choosing $g(\cdot) = \| \cdot \|_{\mathcal{X}}^2$.

To solve these types of problems, one popular approach includes penalty based methods which are described as differential inclusion

$$\dot{x}(t) + A(x(t)) + \beta(t)\partial f(x(t)) \ni 0 \tag{24}$$

where the penalty parameter $\beta(t)$ tends to infinity. As demonstrated in [9], when the monotone operator is a sub-differential $A = \partial g$, then we may equivalently consider the differential inclusion

$$\dot{x}(t) + \lambda(t)\partial g(x(t)) + \partial f(x(t)) \ni 0$$

with vanishing parameter $\lambda(t)$. In summary, analyses of the above differential inclusion can be translated to the differential equation

$$\dot{x}(t) + \nabla f(x(t)) + \lambda(t)x(t) = 0 \tag{25}$$

describing the regularized steepest descent in continuous time. Note that reg-GD defined in (3) can be interpreted as explicit Euler discretization of (25).

### D.2.1 Related work in the deterministic setting

The analysis of dynamical systems corresponding to (24) with $\partial f = 0$ dates back to the 1970s. For instance, in [13], it was shown that for $A = \partial g$, where $g$ is lower semicontinuous, proper, and convex, the trajectory converges weakly to a minimizer of $g$. More generally, for maximal monotone operators $A$, the ergodic average of the trajectory converges weakly to a point in $A^{-1}(\{0\})$ [23].

The penalty-based differential inclusion (24) was introduced in [9], where the authors established weak ergodic convergence (and even strong convergence for strongly monotone operators $A$) under the integrability condition

$$\int_0^\infty \beta(t) \left[ \Psi^*\left(\frac{p}{\beta(t)}\right) - \sigma_C\left(\frac{p}{\beta(t)}\right) \right] \mathrm{d}t < \infty \quad \text{for all } p \in \mathrm{range}(N_C),$$

where $\Psi^*$ denotes the Fenchel conjugate of $\Psi$, and $\sigma_C$ is the support function of the set $C$. This condition is now commonly referred to as the *Attouch–Czarnecki condition*.

Note that a similar condition arises in our analysis as the final requirement in (7). While our condition can be characterized via the Łojasiewicz inequality, the Attouch–Czarnecki condition can be characterized using a quadratic error bound of the form

$$\Psi(x) \geq C \, \mathrm{dist}(x, C)^2,$$

which implies that

$$\Psi^*(p) - \sigma_C(p) \leq \frac{\|p\|^2}{2C},$$

see for instance [9, 11] for more details. In this case, the Attouch–Czarnecki condition is guaranteed under integrability conditions on the penalty function $\beta(\cdot)$.

In the discrete-time setting, the Attouch–Czarnecki condition translates into a summability condition involving both the penalty sequence and the step-sizes. For instance, [60] introduces a coupled gradient method with exterior penalization, leading to the condition

$$\sum_{n \in \mathbb{N}} \alpha_n \beta_n \left[ \Psi^* \left( \frac{p}{\beta_n} \right) - \sigma_C \left( \frac{p}{\beta_n} \right) \right] < \infty \,.$$

Here, the author considers the case where $A = \nabla g$ and both $\nabla f$ and $\nabla g$ are Lipschitz continuous, establishing weak convergence under convexity of $g$, and strong convergence when $g$ is strongly convex. Other results in the discrete-time setting include splitting-based discretization schemes [10, 11, 27] whose convergence analysis rely on some similar variant of the Attouch–Czarnecki condition.

### D.2.2 Convergence rate for reg-GD

In the following, we quantify the rate of convergence of reg-GD defined in (3). Our derived rates are consistent with known results for the ODE (25). In particular, for sufficiently small step-sizes, i.e. $\alpha_k \leq \frac{2}{L}$, our results match those derived in [7, Theorem 5] when defining the numerical time $t_k = \sum_{i=1}^{k} \alpha_i$ for $k \in \mathbb{N}$ and noting that $t_k \sim \frac{C_\alpha}{1-q} k^{1-q}$ and $\lambda_k \sim C_\lambda (\frac{1-q}{C_\alpha} t_k)^{-p/(1-q)}$ for $q < 1$. The proof follows the strategy of [7, Theorem 5].

**Theorem D.5.** *Suppose that Assumption 1.1 is satisfied. Let $C_\alpha, C_\lambda > 0$, $p \in (0,1]$ and $q \in [0, 1-p]$. Let $(X_k)_{k \in \mathbb{N}_0}$ be generated by (3) for all $k \in \mathbb{N}$, $(\lambda_k)_{k \in \mathbb{N}} = (C_\lambda k^{-p})_{k \in \mathbb{N}}$ and $(\alpha_k)_{k \in \mathbb{N}} = (C_\alpha k^{-q})_{k \in \mathbb{N}}$ such that the following conditions are satisfied:*

$$\begin{cases} C_\alpha < \frac{2}{L} & : q = 0 \\ 2 C_\lambda C_\alpha > 1 - q & : q = 1 - p \text{ and } q \neq 0 \\ 2 C_\lambda C_\alpha (1 - \frac{L C_\alpha}{2}) > 1 & : q = 0 \text{ and } p = 1 \end{cases}.$$

*Then it holds that*

    *(i)* $E_k \in \mathcal{O}(k^{-1+q})$,

    *(ii)* $f(X_k) - f(x_*) \in \mathcal{O}(k^{-p})$,

    *(iii)* $\|X_k - x_{\lambda_{k+1}}\|_{\mathcal{X}}^2 \in \mathcal{O}(k^{-1+q+p})$ *for* $q \in [0, 1-p)$, *and*

    *(iv)* $\lim_{k \to \infty} \|X_k - x_*\|_{\mathcal{X}} = 0$ *for* $q \in [0, 1-p)$.

*Proof.* (i): Proposition D.1 with $D_k \equiv 0$ guarantees that, for all $k \in \mathbb{N}_0$ with $\alpha_k \leq \frac{2}{L+\lambda_k}$, one has

$$E_k \leq \left( 1 - 2\lambda_k \alpha_k \left( 1 - \frac{L + \lambda_k}{2} \alpha_k \right) \right) E_{k-1} + \frac{\lambda_k - \lambda_{k+1}}{2} \|x_*\|_{\mathcal{X}}^2 \,.$$

Set $\beta = 1 - q$ and for $k \in \mathbb{N}$ define[1] $\varphi_k = E_k k^\beta$. By assumption on $(\alpha_k)_{k \in \mathbb{N}}$ one has $\alpha_k < \frac{2}{L}$ for all but finitely many indices $k$. Therefore there exists an $N \in \mathbb{N}$ such that for all $k > N$

$$\varphi_k \leq \left( 1 - 2\lambda_k \alpha_k \left( 1 - \frac{L + \lambda_k}{2} \alpha_k \right) \right) \frac{k^\beta}{(k-1)^\beta} \varphi_{k-1} + \frac{\lambda_k - \lambda_{k+1}}{2 k^{-\beta}} \|x_*\|_{\mathcal{X}}^2 \,. \tag{26}$$

Using Lemma B.4, one has

$$\frac{\lambda_k - \lambda_{k+1}}{2 k^{-\beta}} \leq \frac{C_\lambda p}{2} k^{\beta - 1 - p}$$

and there exist $\varepsilon, \varepsilon' > 0$ such that after possibly increasing $N$ one has for all $k \geq N$

$$\left( 1 - 2\lambda_k \alpha_k \left( 1 - \frac{L + \lambda_k}{2} \alpha_k \right) \right) \frac{k^\beta}{(k-1)^\beta} \leq \left( 1 - 2\lambda_k \alpha_k \left( 1 - \frac{L + \lambda_k}{2} \alpha_k \right) \right) \left( 1 + \frac{(\beta + \varepsilon)}{k} \right)$$

$$\leq 1 - \varepsilon' \lambda_k \alpha_k, \tag{27}$$

---

[1] In order to avoid confusion, we note that $\varphi_k$ is defined differently as in the proofs of Theorem D.3 and Theorem D.4

where in the case $q = 1 - p$ and $q \neq 0$ we have used that $2C_\lambda C_\alpha > 1 - q$ and in the case $q = 0$ and $p = 1$ we have used that $2C_\lambda C_\alpha(1 - \frac{LC_\alpha}{2}) > 1$. Inserting these inequalities in (26),

$$\varphi_k \leq (1 - \varepsilon' \lambda_k \alpha_k)\varphi_{k-1} + \frac{C_\lambda p}{2} k^{\beta - 1 - p} \|x_*\|_{\mathcal{X}}^2$$

$$= (1 - \varepsilon' C_\lambda C_\alpha k^{-p-q})\varphi_{k-1} + \frac{C_\lambda p}{2} k^{-p-q} \|x_*\|_{\mathcal{X}}^2,$$

which is equivalent to

$$\left(\varphi_k - \frac{p}{2\varepsilon' C_\alpha} \|x_*\|_{\mathcal{X}}\right) \leq (1 - \varepsilon' C_\lambda C_\alpha k^{-p-q})\left(\varphi_{k-1} - \frac{p}{2\varepsilon' C_\alpha} \|x_*\|_{\mathcal{X}}\right).$$

Therefore, by induction we get

$$\left(\varphi_k - \frac{p}{2\varepsilon' C_\alpha} \|x_*\|_{\mathcal{X}}\right) \leq \left(\varphi_N - \frac{p}{2\varepsilon' C_\alpha} \|x_*\|_{\mathcal{X}}\right) \prod_{i=N+1}^{k} (1 - \varepsilon' C_\lambda C_\alpha (i+1)^{-p-q})$$

$$\leq \left(\varphi_N - \frac{p}{2\varepsilon' C_\alpha} \|x_*\|_{\mathcal{X}}\right) \exp\left(-\sum_{i=N+1}^{k} \varepsilon' C_\lambda C_\alpha (i+1)^{-p-q}\right) \xrightarrow{k \to \infty} 0,$$

where convergence holds since $p + q \leq 1$. This implies

$$\limsup_{k \to \infty} \varphi_k = \limsup_{k \to \infty} E_k k^\beta \leq \frac{p}{2\varepsilon' C_\alpha} \|x_*\|_{\mathcal{X}}.$$

(ii): Follows from (i) and Lemma D.2, using that $p \leq 1 - q$.

(iii): Follows from (i) and Lemma D.2.

(iv): Follows from (iii) together with $\lim_{\lambda \to 0} \|x_\lambda - x_*\|_{\mathcal{X}} = 0$. $\qquad \square$

In the spirit of Section 2.3, we will derive optimal decay rates for the step-size and regularization decay for the convergence to the minimum-norm solution under the additional assumption that there exist $C_{\text{reg}}, \xi > 0$ with

$$\|x_\lambda - x_*\|_{\mathcal{X}} \leq C_{\text{reg}} \lambda^\xi, \quad \lambda \in (0, 1]$$

see also Section 3. Using Theorem D.5, one has

$$\|X_k - x_*\|_{\mathcal{X}}^2 = \mathcal{O}(k^{-\min(1-q-p, 2\xi p)}). \tag{28}$$

Thus, we get the optimal rate of convergence for $C_\alpha < \frac{2}{L}, q = 0$ and $p = \frac{1}{2\xi + 1}$, which gives

$$\|X_k - x_*\|_{\mathcal{X}}^2 \in \mathcal{O}(k^{-\frac{2\xi}{2\xi+1}}).$$

In Figure 14, we illustrate the convergence rate on depending on the decay-rates $p, q$ for $\xi = \frac{1}{4}$.

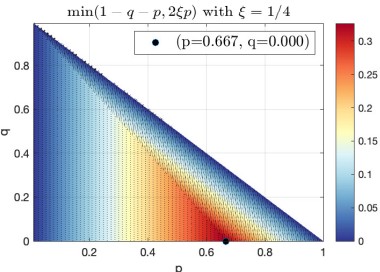

Figure 14: Convergence rate for $\|X_k - x_*\|_{\mathcal{X}}^2$ in the situation of Theorem D.5 under the Polyak-Łojasiewicz inequality, i.e. under (10) with $\xi = \frac{1}{4}$.

## D.3 $L^2$-convergence rate for reg-SGD: Proof of Theorem 2.3

In the following, we formulate Theorem 2.3 in more details and provide a full prove.

**Theorem D.6.** *Suppose that Assumption 1.1 and Assumption 1.2 are satisfied. Let $C_\alpha, C_\lambda > 0$, $p \in (0, \frac{1}{2}]$ and $q \in (p, 1-p]$. Let $(X_k)_{k \in \mathbb{N}_0}$ be generated by (4) with $(\alpha_k)_{k \in \mathbb{N}} = (C_\alpha k^{-q})_{k \in \mathbb{N}}$ and $(\lambda_k)_{k \in \mathbb{N}} = (C_\lambda k^{-p})_{k \in \mathbb{N}}$. If $q = 1 - p$ we additionally assume that $2C_\lambda C_\alpha > 1 - q$. Then, it holds that $\lim_{k \to \infty} \mathbb{E}[\|X_k - x_*\|_{\mathcal{X}}^2] = 0$ and*

*(i) $\mathbb{E}[E_k] \in \mathcal{O}(k^{-\min(1-q, q-p)})$,*

*(ii) $\mathbb{E}[f(X_k) - f(x_*)] \in \mathcal{O}(k^{-\min(p, q-p)})$, and*

*(iii) $\mathbb{E}[\|X_k - x_{\lambda_{k+1}}\|_{\mathcal{X}}^2] \in \mathcal{O}(k^{-\min(1-q-p, q-2p)})$ for $p \in (0, \frac{1}{3})$ and $q \in (2p, 1-p)$.*

*Proof.* We will only prove the first claim (i). Due to Theorem 2.2, one has $\mathbb{E}[\|X_k - x_*\|_{\mathcal{X}}^2] \to 0$ if $q > p$ and $p + q < 1$. The statements (ii)-(iii) follow analogously to the proof of Theorem D.5. Let $\beta = \min(1 - q, q - p)$ and $\varphi_k = \mathbb{E}[E_k]k^\beta$. By assumption on $(\alpha_k)_{k \in \mathbb{N}}$ one has $\alpha_k < \frac{2}{L}$ for all but finitely many $k \in \mathbb{N}$ so that, using Proposition D.1,

$$\varphi_k \leq \left(1 - 2\lambda_k \alpha_k \left(1 - \frac{L + \lambda_k}{2}\alpha_k\right) + \frac{L + \lambda_k}{2}\alpha_k^2 A\right)\frac{k^\beta}{(k-1)^\beta}\varphi_{k-1}$$
$$+ \frac{\lambda_k - \lambda_{k+1}}{2k^{-\beta}}\|x_*\|_{\mathcal{X}}^2 + \frac{L + \lambda_k}{2k^{-\beta}}\alpha_k^2\left(A\frac{\lambda_k}{2}\|x_*\|_{\mathcal{X}}^2 + C\right).$$

Since $q > p$, $p + q \geq 1$, and $2C_\lambda C_\alpha > \min(p, 1 - 2p)$ in the case that $q = 1 - p$ one can show as in (27) that there exist $\varepsilon' > 0$ and $N \in \mathbb{N}$ such that for all $k \geq N$

$$\varphi_k \leq (1 - \varepsilon' \lambda_k \alpha_k)\varphi_{k-1} + \frac{C_\lambda p}{2}k^{\beta-1-p}\|x_*\|_{\mathcal{X}}^2 + \frac{L + \varepsilon'}{2}C_\alpha^2\left(\frac{A\varepsilon'}{2}\|x_*\|_{\mathcal{X}}^2 + C\right)k^{-2q+\beta}. \quad (29)$$

By choice of $\beta$, one has $p + q = \max(1 + p - \beta, 2q - \beta)$. Therefore, we can show analogously to the proof of Theorem D.5 that

$$\limsup_{k \to \infty} \varphi_k = \limsup_{k \to \infty} \mathbb{E}[E_k]k^\beta < \infty.$$

$\square$

## D.4 Almost sure convergence rate for reg-SGD: Proof of Theorem 2.4

In the following, we formulate Theorem 2.4 in more details and provide a full prove. The proof of the almost sure convergence rates requires a sophisticated application of the Robbins-Siegmund theorem, Corollary B.6. For this, we use the variation of constants formula to separate the influence of the stochastic noise term $(D_k)_{k \in \mathbb{N}}$ and the deterministic change in the global minimum of the regularized objective function $(x_{\lambda_k} - x_{\lambda_{k+1}})_{k \in \mathbb{N}}$.

**Theorem D.7.** *Suppose that Assumption 1.1 and Assumption 1.2 are satisfied. Let $C_\alpha, C_\lambda > 0$, $p \in (0, \frac{1}{2})$ and $q \in (\frac{1}{2}, 1 - p]$. Let $(X_k)_{k \in \mathbb{N}_0}$ be generated by (4) with $(\alpha_k)_{k \in \mathbb{N}} = (C_\alpha k^{-q})_{k \in \mathbb{N}}$ and $(\lambda_k)_{k \in \mathbb{N}} = (C_\lambda k^{-p})_{k \in \mathbb{N}}$. Let $\beta \in (0, 2q - 1)$ and, if $q = 1 - p$, we assume that $2C_\lambda C_\alpha > \min(\beta, 1 - q)$. Then,*

*(i) $E_k \in \mathcal{O}(k^{-\min(\beta, 1-q)})$ almost surely,*

*(ii) $f(X_k) - f(x_*) \in \mathcal{O}(k^{-\min(\beta, p)})$ almost surely,*

*(iii) $\|X_k - x_{\lambda_{k+1}}\|_{\mathcal{X}} \in \mathcal{O}(k^{-\min(\beta-p, 1-q-p)})$ almost surely, and*

*(iv) $\lim_{k \to \infty} \|X_k - x_*\|_{\mathcal{X}} \to 0$ almost surely for $p \in (0, \frac{1}{3})$ and $q \in (\frac{p+1}{2}, 1 - p)$.*

*Proof.* We will only prove property (i). Properties (ii)-(iv) follow analogously to the proof of Theorem D.5. By Proposition D.1, for all $k \in \mathbb{N}$ with $\alpha_k \leq \frac{2}{L+\lambda_k}$ one has

$$\mathbb{E}[E_k \mid \mathcal{F}_{k-1}] \leq \left(1 - 2\lambda_k\alpha_k\left(1 - \frac{L+\lambda_k}{2}\alpha_k\right) + \frac{L+\lambda_k}{2}\alpha_k^2 A\right)E_{k-1}$$
$$+ \frac{\lambda_k - \lambda_{k+1}}{2}\|x_*\|_{\mathcal{X}}^2 + \frac{L+\lambda_k}{2}\alpha_k^2\left(A\frac{\lambda_k}{2}\|x_*\|_{\mathcal{X}}^2 + C\right).$$

For $k \in \mathbb{N}_0$ we define

$$\Psi_k = \sum_{i=1}^{k}\frac{\lambda_i - \lambda_{i+1}}{2}\|x_*\|_{\mathcal{X}}^2 \prod_{j=i+1}^{k}\left(1 - 2\lambda_j\alpha_j\left(1 - \frac{L+\lambda_j}{2}\alpha_j\right) + \frac{L+\lambda_j}{2}\alpha_j^2 A\right)$$

and $\tilde{E}_k = E_k - \Psi_k$. Since $\alpha_k \to 0$, one has for all but finitely many $k$'s that

$$\mathbb{E}[\tilde{E}_k \mid \mathcal{F}_{k-1}] = \mathbb{E}[E_k - \Psi_k \mid \mathcal{F}_{k-1}]$$
$$\leq \left(1 - 2\lambda_k\alpha_k\left(1 - \frac{L+\lambda_k}{2}\alpha_k\right) + \frac{L+\lambda_k}{2}\alpha_k^2 A\right)E_{k-1}$$
$$+ \frac{\lambda_k - \lambda_{k+1}}{2}\|x_*\|_{\mathcal{X}}^2 + \frac{L+\lambda_k}{2}\alpha_k^2\left(A\frac{\lambda_k}{2}\|x_*\|_{\mathcal{X}}^2 + C\right) - \Psi_k$$
$$= \left(1 - 2\lambda_k\alpha_k\left(1 - \frac{L+\lambda_k}{2}\alpha_k\right) + \frac{L+\lambda_k}{2}\alpha_k^2 A\right)E_{k-1}$$
$$+ \frac{\lambda_k - \lambda_{k+1}}{2}\|x_*\|_{\mathcal{X}}^2 + \frac{L+\lambda_k}{2}\alpha_k^2\left(A\frac{\lambda_k}{2}\|x_*\|_{\mathcal{X}}^2 + C\right)$$
$$- \left(1 - 2\lambda_k\alpha_k\left(1 - \frac{L+\lambda_k}{2}\alpha_k\right) + \frac{L+\lambda_k}{2}\alpha_k^2 A\right)\Psi_{k-1} - \frac{\lambda_k - \lambda_{k+1}}{2}\|x_*\|_{\mathcal{X}}^2$$
$$= \left(1 - 2\lambda_k\alpha_k\left(1 - \frac{L+\lambda_k}{2}\alpha_k\right) + \frac{L+\lambda_k}{2}\alpha_k^2 A\right)\tilde{E}_{k-1}$$
$$+ \frac{L+\lambda_k}{2}\alpha_k^2\left(A\frac{\lambda_k}{2}\|x_*\|_{\mathcal{X}}^2 + C\right).$$

Let $\beta \in (0, 2q-1)$, $\tilde{\beta} = \min(\beta, 1-q)$ and $\varphi_k = \tilde{E}_k k^{\tilde{\beta}}$. By assumption on $(\alpha_k)_{k \in \mathbb{N}}$ one has $\alpha_k < \frac{2}{L}$ for all but finitely many $k \in \mathbb{N}$ so that

$$\mathbb{E}[\varphi_k \mid \mathcal{F}_{k-1}] \leq \left(1 - 2\lambda_k\alpha_k\left(1 - \frac{L+\lambda_k}{2}\alpha_k\right) + \frac{L+\lambda_k}{2}\alpha_k^2 A\right)\frac{k^{\tilde{\beta}}}{(k-1)^{\tilde{\beta}}}\varphi_k$$
$$+ \frac{L+\lambda_k}{2}C_\alpha^2\left(A\frac{\lambda_k}{2}\|x_*\|_{\mathcal{X}}^2 + C\right)k^{-2q+\beta}$$

Since $q > p$, $p + q \geq 1$, and $2C_\lambda C_\alpha > \tilde{\beta}$ in the case that $q = 1 - p$, one can show as in (27) that there exist $\varepsilon' > 0$ and $N \in \mathbb{N}$ such that for all $k \geq N$

$$\mathbb{E}[\varphi_k \mid \mathcal{F}_{k-1}] \leq (1 - \varepsilon'\lambda_k\alpha_k)\varphi_{k-1} + \frac{L+\varepsilon'}{2}C_\alpha^2\left(\frac{A\varepsilon'}{2}\|x_*\|_{\mathcal{X}}^2 + C\right)k^{-2q+\beta}, \qquad (30)$$

for all sufficiently large $k$.

In order to apply the Robbins-Siegmund theorem, Corollary B.6, we first prove that $(\Psi_k)_{k \in \mathbb{N}} \in \mathcal{O}(k^{-\tilde{\beta}})$. Note that $(\Psi_k)_{k \in \mathbb{N}}$ is a deterministic sequence that satisfies for all $k \in \mathbb{N}$

$$\Psi_k = \left(1 - 2\lambda_k\alpha_k\left(1 - \frac{L+\lambda_k}{2}\alpha_k\right) + \frac{L+\lambda_k}{2}\alpha_k^2 A\right)\Psi_{k-1} + \frac{\lambda_k - \lambda_{k+1}}{2}\|x_*\|_{\mathcal{X}}^2.$$

Therefore, analogously to the proof in the deterministic setting, see Theorem D.5 and especially (27), we get $\Psi_k \in \mathcal{O}(k^{-\tilde{\beta}})$, i.e. $(\Psi_k k^{\tilde{\beta}})_{k \in \mathbb{N}}$ is bounded and, subsequently, $(\varphi_k)_{k \in \mathbb{N}}$ is uniformly bounded from below. Now, since $\beta < 2q - 1$ we get $\sum k^{-2q+\beta} < \infty$. Hence, we can apply Corollary B.6 to get almost sure convergence of $(\varphi_k)_{k \in \mathbb{N}}$ and, thus, $\tilde{E}_k \in \mathcal{O}(k^{-\tilde{\beta}})$ almost surely. Together with $\tilde{E}_k = E_k - \Psi_k$ and $\Psi_k = \mathcal{O}(k^{-\tilde{\beta}})$, this implies that $E_k \in \mathcal{O}(k^{-\tilde{\beta}})$ almost surely. $\qquad\square$

# E   Properties of the Tikhonov regularization

In the following section, we want to describe scenarios in which (10) is satisfied.

**Linear inverse problems.**  Let $A : \mathcal{X} \to \mathcal{Y}$ be a compact linear operator between two Hilbert spaces. For $y \in \mathcal{R}(A) \oplus \mathcal{R}(A)^{\perp}$, the minimum-norm solution to the problem

$$\min_{x \in \mathcal{X}} f(x), \quad f(x) = \frac{1}{2}\|Ax - y\|_{\mathcal{Y}}^2$$

can be written in the form of the singular value decomposition (SVD) of $A$:

$$x_* = A^{\dagger}y = \sum_{n \in \mathbb{N}} \frac{1}{\sigma_n}\langle y, u_n\rangle_{\mathcal{Y}} v_n,$$

where $(\sigma_n, u_n, v_n)_{n \in \mathbb{N}}$ is the SVD of $A$ with singular values $(\sigma_n)_{n \in \mathbb{N}}$, an orthonormal basis $(u_n)_{n \in \mathbb{N}}$ of $\overline{\mathcal{R}(A)}$, and an orthonormal basis $(v_n)_{n \in \mathbb{N}}$ of $\overline{\mathcal{R}(A^*)}$. Similarly, for any $\lambda > 0$, the unique minimizer of

$$\min_{x \in \mathcal{X}} f_\lambda(x), \quad f_\lambda(x) = \frac{1}{2}\|Ax - y\|_{\mathcal{Y}}^2 + \frac{\lambda}{2}\|x\|_{\mathcal{X}}^2$$

can also be written using the SVD as:

$$x_\lambda = \sum_{n \in \mathbb{N}} \frac{\sigma_n}{\sigma_n^2 + \lambda}\langle y, u_n\rangle_{\mathcal{Y}} v_n.$$

To obtain a convergence rate for $\|x_* - x_\lambda\|_{\mathcal{X}}$ as $\lambda \to 0$, we need to bound

$$r_n(\lambda) := \frac{1}{\sigma_n} - \frac{\sigma_n}{\sigma_n^2 + \lambda} = \frac{\sigma_n(\sigma_n^2 + \lambda) - \sigma_n^3}{\sigma_n^2(\sigma_n^2 + \lambda)} = \frac{\lambda}{\sigma_n(\sigma_n^2 + \lambda)}.$$

However, when $A$ is infinite-dimensional, the singular values $\sigma_n$ are positive and satisfy $\lim_{n \to \infty} \sigma_n = 0$, meaning that $r_n(\lambda)$ remains unbounded. Therefore, without additional assumptions, we can only deduce that

$$\lim_{\lambda \to 0} \|x_* - x_\lambda\|_{\mathcal{X}} = 0,$$

but without a specific rate in $\lambda$. To impose a convergence rate, one typically assumes a so-called source condition [31] common in the inverse problem literature, which imposes a smoothness assumption on the (infinite-dimensional) minimum-norm solution $x_*$.

In terms of the SVD, the source condition with parameter $\nu > 0$ can be described by the representation of the minimum-norm solution

$$A^{\dagger}y = x_* = \sum_{n \in \mathbb{N}} \sigma_n^{\nu}\langle w, v_n\rangle_{\mathcal{X}} v_n,$$

for some bounded $w \in \mathcal{X}$. Using this representation together with the SVD expression for $x_\lambda$, one can derive the following bound for the error $\|x_* - x_\lambda\|_{\mathcal{X}}^2$:

$$\|x_* - x_\lambda\|_{\mathcal{X}}^2 \leq \begin{cases} C_\nu \lambda^2, & \nu \geq 2, \\ C_\nu \lambda^\nu, & \nu < 2, \end{cases}$$

where $C_\nu$ is a constant depending on $\nu > 0$.

**Łojasiewicz condition.**  Introduced in the 1960s by Łojasiewicz [54, 53], the Łojasiewicz inequality (31) has become one of the standard assumptions for convergence of gradient based algorithms [55, 1, 69, 28, 72]. It has the appeal that it is locally satisfied by every analytic objective function [54]. In the machine learning community, (31) with $\tau = \frac{1}{2}$ is especially popular, since it allows linear convergence of deterministic algorithms in non-convex situations [48, 73]. We cite a recent result in [56] that derives and upper bound for the distance of $x_\lambda$ and $x_*$ under validity of the Łojasiewicz inequality. The result uses a connection between the Łojasiewicz inequality and a Hölderian error bound derived in [19].

**Lemma E.1** (See Theorem 5 in [19] and 4.7 in [56])**.** *Let $f : \mathcal{X} \to \mathbb{R}$ be a differentiable, convex function with* $\arg\min f \neq \emptyset$.

*(i) Assume that there exist $\tilde{C}, r > 0$ and $\tau \in [0, 1)$ such that*

$$(f(x) - f(x_*))^\tau \leq \tilde{C} \|\nabla f(x)\|_{\mathcal{X}} \quad \text{for all } x \in f^{-1}([f(x_*), f(x_*) + r]). \quad (31)$$

*Then there exists a constant $C' > 0$ such that with $\rho = \frac{1}{1-\tau}$ it holds that*

$$f(x) - f(x_*) \geq C' \inf_{\hat{x} \in \arg\min f} \|x - \hat{x}\|_{\mathcal{X}}^\rho \quad \text{for all } x \in f^{-1}([f(x_*), f(x_*) + r]). \quad (32)$$

*(ii) Assume that (32) holds. Then, there exist $C_{reg}, \varepsilon > 0$ such that*

$$\|x_\lambda - x_*\|_{\mathcal{X}} \leq C_{reg} \lambda^{\frac{1}{2\rho}} \quad \text{for all } \lambda \in [0, \varepsilon].$$

Finally, we note that in linear inverse problems a Łojasiewicz condition can be verified under the source condition discussed before, see [35, Theorem 5.10].

