# OpenReview forum: "Controlling the Flow: Stability and Convergence for Stochastic Gradient Descent with Decaying Regularization"
_NeurIPS.cc/2025/Conference — NeurIPS 2025 poster_

### Official Review · Reviewer_25Se · 2025-06-22

**Clarity:** 4
**Significance:** 3
**Originality:** 3
**Rating:** 5
**Confidence:** 4

**Summary:**

This paper studies the stochastic gradient descent algorithm with Tikhonov regularization (reg-SGD), applied to the minimization of a convex smooth function on a Hilbert space. Theoretical results show that the iterates of the reg-SGD algorithm converge in norm, almost surely, to the minimum-norm solution with a careful choice of the step size and regularization parameters and under a certain condition on the minimum norm solution itself. When the latter assumption cannot be verified based on problem-specific structures, such as the source condition or the Łojasiewicz inequality, a second result on convergence in norm in the mean squared sense may still hold with more lenient assumptions. The authors offer both a theoretical analysis and a practical guidance for the optimal scheduling of the decaying step size and regularization parameters. Practical importance and feasibility of the algorithm are demonstrated on a Radon transform inversion arising in X-ray tomography problems.

**Questions:**

1. As elaborated in the Weaknesses Section, I suggest drawing concrete connections to the effects of minibatch sampling throughout the work. Specifically, this may be done through suggesting how **Assumption 1.2** may be verifiable and including the minibatch size as a tuning parameter in **Section 3.2**. Space constraints may be overcome by relegating them to the Supplements.
2. How pliable do you believe the analysis of the method is to the following two extensions: (i) problems with *noise*, where a minimum-norm solution may not be optimal, as acknowledged in Section 4; and, (ii) problems with a *nonlinear operator*, which naturally arise in PDE parameter identification problems? It is conceivable both that the noise is also present in (ii) and that the need for SGD arises from a large number of independent observations.
3. Section 1.3 leaves somewhat open the connection between this work and the parallel line of investigation into the generalizability of "flat minima" (and whether SGD can reliably find it). Can the analysis given in this work shed a light on a better understanding of when the SGD-found optimal are generalizable? Is the suggested generalizability in machine learning community due to the SGD "imitating" the effects of Tikhonov regularization, and if so, should the introduction of reg-SGD imply that the engineers revise their existing scheduling of parameter decay and early stopping rules to prevent "excessive" regularization?

**Ethical Concerns:**

["NO or VERY MINOR ethics concerns only"]

**Final Justification:**

Weakness and Questions raised by the current reviewer were satisfactorily addressed by the authors. I suggest the promised additions and commentaries regarding Weakness 1 and Question 2 to be incorporated in the revision.

**Limitations:**

Yes.

**Paper Formatting Concerns:**

None.

**Quality:**

4

**Strengths And Weaknesses:**

* Strengths
  1. The problem is clearly motivated, both theoretically and in practical examples and the analysis of the regularized SGD algorithm is technically solid with no blatant flaws.
  2. Extension of the by now various results on SGD algorithm to Hilbert spaces and beyond is an interesting problem in and of itself in applied mathematics. The authors have made a contribution towards connecting the existing machine learning point of view towards SGD algorithms and the existing, albeit somewhat detached, literature on the analysis of Tikhonov regularization.

* Weaknesses
  1. The current exposition seems to only analyze SGD "at large" without delving into the specific ways in which the existing SGD algorithms are implemented. More specifically: in many machine learning tasks, and also in the tomography problem investigated in Section 3.2, the need for SGD often arises from the large number of observations, presumably "independent" in a precisely probabilistic or looser sense, hence the replacement of the full-scan gradient evaluation with a minibatch gradient evaluation. There are at least two sections in this work that may benefit from a more concrete connection to the assumption of gradient noise arising from minibatch sampling: **Assumption 1.2**, which I believe will be verifiable in this case; and the empirical investigations in **Section 3.2**, where the trade-offs involved in minibatch size and the performance of the reg-SGD remain unexplored.

---

> ### Author Rebuttal · Authors · 2025-07-31
>
> Dear reviewer,
>
> We sincerely thank you for the time spent evaluating our work and your overall positive feedback.
>
> # W1/Q1. Incorporation of mini-batch gradient evaluation:
>
> This is an important point that we would like to incorporate into the revised version of our manuscript. When introducing a mini-batch scheme, one can easily verify Assumption 1.2. Since the sample across the mini-batch is usually assumed to be i.i.d., we can simply apply Bienaymé's identity to show a variance reduction and, therefore, the verification of the ABC condition. Another interesting line of study would be the derivation of feasible step sizes and regularization parameters when the mini-batch size is increased dynamically.
>
> ## Action:
> In our revised manuscript, we will add a discussion about empirical risk minimization and the incorporation of a mini-batch sampling scheme. Moreover, we will discuss the verification of Assumption 1.2 within this setting.
>
> # Q2. Possible extensions
> ## Q2.1. Noisy observations:
> The extension of our analysis to the noisy observation setting is planned for future work. In particular, there are several papers on the application of SGD in the context of linear inverse problems, where the least square problem is viewed as finite sum loss. See for instance:
> - "On the regularizing property of stochastic gradient descent"
> - "On the discrepancy principle for stochastic gradient descent"
> - "On the saturation phenomenon of stochastic gradient descent for linear inverse problems"
>
> The analysis for vanilla SGD is usually based on the fact that in the finite sum setting the iterates remain in the range of the adjoint of the forward operator. In a noisy operator evaluation setup this property might be violated. Using our reg-SGD analysis we will be able to derive a-priori and even a-posteriori stopping rules to ensure convergence as classical regularization method.
>
> ## Q2.2. Nonlinear models:
>
> Similarly as the extension to the noisy observation setting, we are planning to extend our analysis to the non-convex setting in the context of non-linear inverse problems. There has been much work on deterministic iterative regularization methods for non-linear inverse problems which are based on a so-called tangential cone condition, see for instance "Iterative Regularization Methods for Nonlinear Ill-Posed Problems". This condition allows to derive recursive inequalities and to construct suitable Lyapunov functions. More precisely, we anticipate that our analysis can be used to extend the modified Landweber iteration introduced by Scherzer ("A modified Landweber iteration for solving parameter estimation problems") to the stochastic setting. A similar line of work has already investigated the extension of the Landweber iteration to the stochastic setting (which corresponds to vanilla SGD), see "On the convergence of stochastic gradient descent for nonlinear ill-posed problems".
>
> ## Action:
>
> We will add references to the iterative regularization methods applied in linear and non-linear inverse problems. Moreover, we include additional comments on our planned application of reg-SGD in this context.
>
> # Q3. Relation to flat minima
> Thank you for this interesting question. In fact, there seem to be many parallels between Tikhonov regularization, the minimum-norm solution and flat minima with their generalization properties.
> First, using Tikhonov regularization for convex learning problems is known to stabilize solutions, prevent overfitting and can be used to prove PAC learnability of  convex, smooth and bounded learning problems (see Theorem 13.1 and Corollary 13.11 in "Understanding Machine Learning: From Theory to Algorithms").
>
> Another similarity can be drawn by looking at works applying Katzenbergers theorem. In "A Theoretical Framework for Grokking: Interpolation followed by Riemannian Norm Minimisation" the authors show that an appropriate time speedup of gradient flow trajectories leads to a gradient flow on the manifold minimizing the $\ell^2$-norm as $\lambda \to 0$. A similar result for flat minima can be found in "What Happens after SGD Reaches Zero Loss? –A Mathematical Framework" and "Singular-limit analysis of gradient descent with noise injection" where it is shown that SGD (after an appropriate time change) leads to an SDE on the manifold of minima that aims to minimize the flatness of the minimum as the learning rate tends to zero.
> It seems to us that the concept of "flat minima" is in some sense similar to the concept of the minimum-norm solution in the context of non-convex optimization.
>
> Finally, we want to point to the connection between introducing Tikhonov regularization and using Dropout (see e.g. "Dropout Regularization Versus l2-Penalization in the Linear Model") and recent findings in "The Surprising Agreement Between Convex Optimization Theory and Learning‑Rate Scheduling for Large Model Training" (ICML 2025).

---

> > ### Comment · Reviewer_25Se · 2025-08-03
> > **Questions Addressed**
> >
> > I thank the authors for their thoughtful response. I am happy with the suggested additions and commentaries in their revision. My scores remain the same.

---

### Official Review · Reviewer_M92n · 2025-06-23

**Clarity:** 4
**Significance:** 3
**Originality:** 3
**Rating:** 5
**Confidence:** 4

**Summary:**

This paper analyzes regularized SGD for the minimization of convex smooth function, with vanishing Tikhonov regularization parameter and proves convergence to the minimum norm solution. The analysis highlights properties about the the flow of SGD and stable learning, how to tune the step sizes and regularization parameters, and is supported by experiments.

**Questions:**

1) The authors might consider moving some of the experiments in Section A.2, which highlight the optimality of the theoretical schedules, to the main body of the paper, perhaps sacrificing one of the real examples of Section 3? I would understand the choice not to do this, but I do feel these appendix experiments are quite important for the point of the paper.
2) Can the authors please do a thorough spelling/grammar check?
3) Empirically, is the method sensitive to the tuned parameters? Can there be some computational savings by using approximations (throughout)?

**Ethical Concerns:**

["NO or VERY MINOR ethics concerns only"]

**Final Justification:**

I still believe this is a solid paper and contribution (especially after the modifications the authors will make), so my rating stays at an Accept.

**Limitations:**

Some limitations are discussed in the conclusion, mostly pointing out future directions.

**Quality:**

3

**Strengths And Weaknesses:**

The paper is well written, easy to follow, and motivates the problem well while highlighting the novelty. The related work is thorough and the theoretical results are clearly stated.

Quick note to check spelling and grammar, I spotted a few errors (e.g. "continuous a line" on line 83).

Section 3.1 felt oddly placed or oddly named. I understand wanting to work through an example later in the paper, but motivating its importance is usually done quite early (I don't have a strong preference just wanted to raise this point).

There are many experiments in the appendices, some of which feel crucial to the main results. It is likely due to space concerns, but perhaps the authors may wish to re-assess which experiments are included in the main body.

---

> ### Author Rebuttal · Authors · 2025-07-30
>
> Dear reviewer,
>
> We sincerely thank you for the time spent evaluating our work and for the positive feedback.
>
> ## W1/Q2. Grammar corrections.
> Thank you for bringing this to our attention. We will carefully proofread the manuscript to correct any grammatical or typographical errors and ensure that the revised version is polished.
>
> ## W2/W3/Q1. Content of Section 3.
> We appreciate your positive feedback on our additional experiments in the appendix. In case of acceptance, we will make use of the additional content page to integrate key experimental results into the main body of our paper. Moreover, we will explore the option of replacing parts of Section 3.1 with more numerical experiments.
>
> ## Q3. Sensitivity to tuning parameters.
> This is a very important question. The only hyperparameters to tune are the initial step size and the initial regularization parameter. We observed that reg-SGD exhibits significantly lower sensitivity to the initial step size compared to vanilla SGD. From our experiments, one can choose even larger initial step sizes for reg‑SGD without encountering the instability that plagues standard SGD. Regarding the regularization parameter, initializing with values that are too small can lead to transient instabilities. Choosing slightly too large initial values for the regularization will be corrected during the iterations by the scheduled decrease.
>
> ### Action.
> We will complement these findings with an additional numerical experiment that systematically compares the three methods from Section 3.2 over a range of initial step sizes and regularization parameters.

---

> > ### Comment · Reviewer_M92n · 2025-08-01
> > **Questions addressed**
> >
> > Thanks for taking my suggestions into account, I am certainly satisfied with the paper and the responses.

---

### Official Review · Reviewer_Ahp1 · 2025-06-29

**Clarity:** 3
**Significance:** 2
**Originality:** 2
**Rating:** 4
**Confidence:** 3

**Summary:**

This paper provide convergence analysis for reg-SGD in optimization problems under convexity and  Łojasiewicz conditions problems. It establishes the $L^2$ and almost-sure convergence of SGD under the ABC conditions of the stochastic noise.

**Questions:**

1. Could you please compare with the following recent work?

Boursier, E., Pesme, S., & Dragomir, R. A. (2025). A Theoretical Framework for Grokking: Interpolation followed by Riemannian Norm Minimisation. arXiv preprint arXiv:2505.20172.

2. Are there any new insights that could potentially be useful be practice according to your theory?

**Ethical Concerns:**

["NO or VERY MINOR ethics concerns only"]

**Final Justification:**

I acknowledge the contribution on the " convergence rates and as a consequence the tuning of step-size vs. decay of regularization" and thus raise the score to 4.

**Limitations:**

See the weakness part.

**Paper Formatting Concerns:**

I do not notice any major formatting issues in this paper.

**Quality:**

2

**Strengths And Weaknesses:**

**Strength** The paper is clear is easy to follow. I believe the result of this paper is correct.

**Weakness**

1. The result is expected and not surprising to me. I think it is well-known that with regularization the algorithm will converge to the min-norm solution, although the existing analysis may not be comprehensive as this paper. I do not think this paper bring new insights to the community.

2. The paper only studies problems under convexity and  Łojasiewicz conditions problems, but it lacks analysis of for general nonconvex optimization problems, which are the most widespread applications of SGD (or reg-SGD). For instance, neural network training.

---

> ### Author Rebuttal · Authors · 2025-07-30
>
> Dear reviewer,
>
> We thank you for the time spent in evaluating our work. In the following, we will clarify your concerns by providing point by point responses to your comments.
>
> ## Weaknesses 1
> You say "The result is expected and not surprising to me. I think it is well-known that with regularization the algorithm will converge to the min-norm solution". It seems like when looking at our article, you completely overlooked the main results - the delicate balance of decaying regularization parameter and step sizes. In particular, our derived convergence rates. We will sharpen the abstract in the final version to hightlight these contributions even more. When considering access to the minimizers $x_\lambda$, convergence to the min-norm solution $x_\ast$ in the sense that $\lVert x_\lambda-x_\ast\rVert \to 0$ is not surprising. However, when considering an iterative regularized optimization scheme it is far from obvious how to schedule the interplay between the regularization parameter and the step-sizes in order to achieve fast convergence to $x_\ast$. See for instance Chapter 5 in "Ill-Posed Problems: Theory and Applications", and in particular Theorem 5.1 and 5.2. In our stochastic setting this involves additional challenges to verify our Theorem 2.1 and 2.2.
>
> Moreover, we emphasize that the **main contribution of the article are the convergence rates** and as a consequence the tuning of step-size vs. decay of regularization. For graphical illustrations please consult our examples. Specifically in Figure 5, you can observe that a too fast decay of the regularization leads to non-convergence.
>
> ## Weakness 2
> First, it should be noted that our main statements only require convexity and not validity of a Lojasiewicz inequality. We use the Lojasiewicz inequality only as an example where $\lVert x_\lambda-x_\ast\rVert$ can be quantified and, thus, optimal hyperparameter schedules can be derived explicitly.
>
> Moreover, it is well-known that Lojasiewicz/gradient domination-assumptions are relevant for the training of neural networks. Gradient domination holds for instance locally for over-parameterized NNs, see for instance Liu et al. "Loss landscapes and optimization in over-parameterized non-linear systems and neural networks". We acknowledge that a comprehensive theoretical analysis in the general non‑convex setting would be highly relevant for understanding deep neural network training. However, before tackling non‑convex problems, it is essential to build a solid foundation by studying the convex case. Insights from convex analysis frequently yield valuable theoretical and practical tools that can be adapted to non‑convex scenarios, as also recently illustrated by the findings in "The Surprising Agreement Between Convex Optimization Theory and Learning‑Rate Scheduling for Large Model Training" (ICML 2025).
>
> ## Question 1
> The work "A Theoretical Framework for Grokking: Interpolation followed by Riemannian Norm Minimisation" analyzes a gradient flow (GF) with constant Tikhonov regularization from a dynamical systems point of view. It considers a family of GF trajectories parametrized by the regularization parameter and shows that, after an appropriate time-speedup, the limiting dynamics as $\lambda \to 0$ is a GF on the manifold of minima that minimizes the $\ell^2$-norm. This work is conceptually different from our analysis. We consider **a single** optimization trajectory with **time-vanishing** Tikhonov regularization. This allows us to prove convergence results, derive convergence rates and present optimal hyperparameter schedules. The work "A Theoretical Framework for Grokking: Interpolation followed by Riemannian Norm Minimisation" considers a **family** of optimization trajectories with **fixed** Tikhonov regularization. Notably,  GF with fixed Tikhonov regularization only finds $x_\lambda \neq x_\ast$. Not only the setting is fundamentally different, but also the statements and implications. While the work "A Theoretical Framework for Grokking: Interpolation followed by Riemannian Norm Minimisation" presents a **qualitative** result, deriving a limiting ODE as $\lambda \to 0$ and $t\to \infty$, one of the main contributions of our work to present **quantitative results** deriving convergence rates and optimal hyperparameter schedules.
>
> ### Action
> We will add a paragraph mentioning  applications of Katzenbergers theorem for optimization schemes in machine learning.
>
> ## Question 2
>  It is non-trivial to choose the decreasing regularization (depending on the step-sizes) to achieve (fast) convergence to the min-norm solution. This has immediate practical relevance, as acknowledged also by other reviewers and demonstrated in our numerical experiments.
>
> In Figure 2,9 and 12 we give a lower bound on the rate of convergence for (S)GD depending on the hyperparameter schedules. In Figure 5 and 8 we show the importance of balancing the regularization decay with the step-size decay.
>
> The main contribution of this work is precisely to quantify the heuristic idea that, as long as the regularization parameter decays sufficiently slow, reg-SGD converges to the minimum-norm solution and to present optimal hyperparameter schedules that can be used in practice to achieve fast convergence to $x_\ast$.

---

### Official Review · Reviewer_icyv · 2025-07-02

**Clarity:** 4
**Significance:** 3
**Originality:** 2
**Rating:** 5
**Confidence:** 5

**Summary:**

# Summary

This paper focuses on an algorithm combining SGD with Tikhonov regularization for minimizing a convex and smooth function $f$.
The paper is mostly theoretical, and provides numerous results showing that the algorithm converges towards the minimal norm minimizer of $f$, thereafter noted $x_*$.

More precisely, the authors leverage the introduction of the regularization term to obtain

- Almost-sure and $L^2$ convergence of the iterates $X_k$ towards $x_*$,
- Convergence of the values $f(X_k)$ towards $\inf f$,
- Convergence rates for the above quantities,

all this under more or less strict assumptions on the parameters of the method (namely: the learning rate and the regularization parameter).

**Questions:**

# Questions/comments

1. I would like the authors to review properly the existing literature on deterministic Tikhonov-gradient algorithms. Then I would like to see some comparison between their results / hypotheses of Section 2.1 with what was done in the deterministic setting, both in the continuous and discrete regime. I believe this would allow the reader to better appreciate the results.

   1. For instance the third assumption in (6) seems strongly connected to assumptions which already appeared (albeit under different forms) in the literature. See

      - p.132 in "Coupling the gradient method with a general exterior penalization scheme for convex minimization"
      - Hypothesis H4 in "Prox-Penalization and Splitting Methods for Constrained Variational Problems"

      and in particular look at how those papers make a connection with  the Łojasiewicz inequality (or other equivalent error bounds such as $f(x) - f(x_*) \geq C dist(x, argmin f)^\tau$ ).

   2. Are the assumptions (7) the same that what is known in the deterministic case? Is there any work able to obtain convergence without making an assumption like in (6)?

   If the above is done properly **I will raise my score**.

2. Appendix D:  It is known that source conditions imply that a Łojasiewicz inequality holds onto some set, see Theorem 5.10 in "Convergence of the forward-backward algorithm: beyond the worst-case with the help of geometry". Can it be proved that the regularization path $x_\lambda$ belongs to such set? If yes, then a connection could be made between the "linear inverse problem" scenario and the "Łojasiewicz" scenario.

3. Authors claim that Tikhonov regularization helps preventing the explosion of the trajectory of SGD. I wonder how true is this claim, given that it is not yet fully known what can or cannot do SGD (here by SGD I mean sampling a function f_i when minimizing an expectation of f_i). For instance recent results (see "Stochastic Gradient Descent Revisited") indicate that SGD does not need bounded trajectories to converge in function values. More precisely, convergence of the values always hold, so under a mere coercivity assumption we can recover the boundedness of the iterates.

4. You certainly want to tell the reader that Assumption 1.2 is not an assumption when $f = \mathbb{E}[f_i]$ and $\hat \nabla f(x) = \nabla f_i(x)$ , and always holds true with $A=4L$ and $C = \mathbb{E}[\Vert \nabla f_i(x_*) \Vert^2]$ .

**Ethical Concerns:**

["NO or VERY MINOR ethics concerns only"]

**Final Justification:**

This is a serious contribution and all the issues I raised have been addressed by the authors.

**Limitations:**

yes

**Paper Formatting Concerns:**

N/A : the paper is seriously written

**Quality:**

4

**Strengths And Weaknesses:**

# Strengths

Overall I was very positively impressed by the paper.

- It is very well written. While the topic is technical, the paper is quite easy to read
- The contribution is very good. The results (see the summary) answer all the questions we could have for such stochastic method (well, one could always ask for high probability results...). The obtained rates are for most of them as good as what is currently known for vanilla SGD.
- The assumptions are minimal, and match what are standard assumptions in the community: no bounded gradient assumption but a mere ABC-like condition (which is always verified for vanilla SGD), basic assumptions on the parameters

# Weaknesses

- The main complaint I have is the connection to the literature. The authors claim that there is one only work on gradient algorithms combining with a Tikhonov term. This is incorrect. The literature on the topic is quite consequent and goes back to the 70's, ranging from simple perturbations of the Landweber algorithm for inverse problems, to general methods for selecting a specific minimizer of a function f. Here are a few entries (there are many others, see the literature therein):
  - "Convergence of diagonally stationary sequences in convex
    optimization"
  - "Coupling forward-backward with penalty schemes and parallel splitting for constrained variational inequalities"
  - "Splitting forward-backward penalty scheme for constrained variational problems"
  - for the applications in inverse problems (that the authors seem to favor):
    - "A modified Landweber iteration for solving parameter
      estimation problems"
    - "Iterative Regularization Methods for Nonlinear Ill-Posed Problems"
    - "Iterative Methods for Approximate Solution of Inverse Problems"
- An other (minor) weakness is that I did not find the numerical experiments very compelling. I would have rather liked to see whether Reg-SGD can help find solutions which generalize better (or no numerics at all).

---

> ### Author Rebuttal · Authors · 2025-07-31
>
> Dear reviewer,
>
> We sincerely thank you for the time spent in evaluating our work and your overall positive feedback. We agree that our connection to the literature of classical deterministic optimization came short and we will make sure to include much more details in our revised manuscript as described below. In the following, we provide point by point responses to your comments. To allow full revision, we give full details how our updated manuscript will look like.
>
> # W1/Q1: Related literature
> Thank you very much for your careful assessment of our literature review and for pointing out important research areas that were previously omitted. We revised Section 1.3 to include additional related work in the areas of constrained optimization and iterative regularization methods. More importantly, we added a detailed discussion of the connection between Tikhonov regularization and constrained convex optimization. We hope these revisions significantly clarify our contributions and better situate our work within the relevant research landscape.
>
> Regarding assumption (7) we now remark that: Our requirements in (7) are very similar to the ones needed in the deterministic setting [Bakushinsky & Goncharsky (1994), Thm 5.1 and Thm 5.2] and are motivated by the corresponding deterministic result in continuous time [Cominetti et al. (2008), Thm 2.2].
>
> ## Action 1 (Updated Sec. 1.3):
>
> We include the following:
>
> **Deterministic Tikhonov regularization** ..... derived in [5]. More generally, differential inclusions for constrained convex optimization problems have been intensively studied in [9, 10, 11, 26, 57]. In App. C.2, we provide more details and illustrate the relation to Tikhonov regularization
> ... Discrete time algorithms with decaying Tikhonov regularization have been analyzed in the context of iterative regularization schemes. For instance, in chapter 5 of [14] iterative regularization is discussed to solve variational inequalities covering (3) for convex $f$ as a special case. Under certain conditions on the regularization decay and the step sizes, strong convergence to the minimum-norm solution can be guaranteed. A related analysis has been considered for non-linear IPs [15]. In the specific application to IPs, (3) is also known as the modified Landweber iteration, where convergence is mainly studied for nonlinear forward models using a-priori and a-posteriori stopping rules  [15, 44, 61]. The theoretical analysis is conducted in a non-convex setting and relies on the so-called tangential cone condition.
>
> ..... Finally, in the context of IPs the regularization properties of (vanilla) SGD have been analyzed for linear forward [39, 41, 43]  and non-linear models [42] based on a-priori and a-posteriori stopping rules.
>
> ## Action 2 (Updated Sec C.2):
>
> We begin Sec C.2 with:
>
> Before discussing the convergence rates for reg-GD, we want to relate our analysis to the literature in convex optimization. For this purpose we formulate our task of finding the minimum-norm solution as constrained optimization problem in form of
> $$ \min_{x\in X} 1/2\lVert x\rVert_{X}^2 \quad \text{s.t.}\quad x\in C:=\arg\min_{y\in X} f(y). $$
> This naturally relates to the task of solving general variational inclusions of form
> $$ 0 \in A(x) + N_C(x) $$
> where $A$ denotes a (maximal) monotone operator and $$N_C(x) = \lbrace v\in X: \langle v,w-x\rangle\le 0\quad \forall w\in C\rbrace$$ is the normal cone of a closed convex set $C$ at $x$. In our setting the operator $A(x)=\nabla_z 1/2\lVert z\rVert_{X}^2\mid_{z=x} = x$ is strongly monotone. Another important class of problems studied in this context are hierarchical optimization problems of finding points in the set
> $$ S = \arg\min \lbrace g(x)\mid x\in\arg\min f(x)\rbrace $$
> for two convex functions $g$ and $f$. This relates to our setting by choosing $g(\cdot) = \lVert \cdot\rVert_{X}^2$.
>
> To solve these types of problems, one popular approach includes penalty based methods which are described as differential inclusion
> $$
>     \dot x(t)+A(x(t))+\beta(t)\partial f(x(t))\ni0 \quad(21)
> $$
> where the penalty parameter $\beta(t)$ tends to infinity. As demonstrated in [9], when the monotone operator is a sub-differential $A = \partial g$, then we may equivalently consider the differential inclusion
> $$
>     \dot x(t)+\lambda(t)\partial g(x(t))+\partial f(x(t))\ni0
> $$
> with vanishing parameter $\lambda(t)$. In summary, analyses of the above differential inclusion can be translated to the differential equation
> $$
>     \dot x(t)+\nabla f(x(t))+\lambda(t)x(t) = 0 \quad (22)
> $$
> describing the regularized steepest descent in continuous time. Note that reg-GD defined in (3) can be interpreted as explicit Euler discretization of (22).
>
> ## Action 3 (additional discussion on related work):
> We added Sec C.2.1:
>
> **C.2.1 Related work**
>
> The analysis of dynamical systems corresponding to (21) with $\partial f = 0$ dates back to the 1970s. For instance, in [13], it was shown that for $A = \partial g$, where $g$ is l.s.c., proper, and convex, the trajectory converges weakly to a minimizer of $g$. More generally, for maximal monotone operators $A$, the ergodic average of the trajectory converges weakly to a point in $A^{-1}(\{0\})$ [22].
>
> The penalty-based differential inclusion (21) was introduced in [9], where the authors established weak ergodic convergence (and even strong convergence for strongly monotone operators $A$) under the integrability condition
> $$
> \int_0^{\infty} \beta(t) \Big[ \Psi^* \big(p/\beta(t)\big) -\sigma_C\big(p/\beta(t)\big) \Big] dt < \infty, \forall p\in range(N_C),
> $$
> where $\Psi^*$ denotes the Fenchel conjugate of $\Psi$, and $\sigma_C$ is the support function of the set $C$. This condition is now commonly referred to as the *Attouch–Czarnecki condition*.
>
> Note that a similar condition arises in our analysis as the final requirement in (6). While our condition can be characterized via the Lojasiewicz inequality, the Attouch–Czarnecki condition can be characterized using a quadratic error bound of the form
> $$
> \Psi(x) \ge C dist(x, C)^2,
> $$
> implying that
> $$
> \Psi^*(p) - \sigma_C(p) \le \|p\|^2/(2C),
> $$
> see for instance [9,11] for more details. In this case, the Attouch–Czarnecki condition is guaranteed under integrability conditions on the penalty function $\beta$.
>
> In the discrete-time setting, the Attouch–Czarnecki condition translates into a summability condition involving both the penalty sequence and the step sizes. For instance, [57] introduces a coupled gradient method with exterior penalization, leading to the condition
> $$
> \sum_{n \in \mathbb{N}} \alpha_n \beta_n \Big[ \Psi^*(p/\beta_n) - \sigma_C(p/\beta_n) \Big] < \infty\,.
> $$
> Here, the author considers the case where $A = \nabla g$ and both $\nabla f$ and $\nabla g$ are Lipschitz continuous, establishing weak convergence under convexity of $g$, and strong convergence when $g$ is strongly convex. Other results in the discrete-time setting include splitting-based discretization schemes [10, 11, 26]  whose convergence analysis rely on some similar variant of the Attouch–Czarnecki condition.
>
> ## Important references
> - [9] Attouch & Czarnecki (2010)
> - [10] Attouch et al. (2012)
> - [11] Attouch et al. (2011)
> - [14] Bakushinsky & Goncharsky (1994)
> - [44] Kaltenbacher (2008)
> - [57] Peypouquet (2012)
> - [61] Scherzer (1998)
>
> # Q2: Relation of source condition and Lojasiewicz condition
> Thank you very much for pointing us to this interesting connection. Assuming that $y\in R(A)\oplus R(A)^\perp$, one can indeed verify that $x_\lambda \in X_{\nu,\delta} = \lbrace (A^* A)^{\nu/2} w\in X\mid w\in X,\lVert w\rVert_{X}\le \delta\rbrace$ due to the SVD representation of $x_\lambda$. More precisely, let $x^\dagger = \sum_{n\in\mathbb N} \sigma_n^\nu \langle w,v_n\rangle_{X} \,  v_n$, then we have
> $$
> x_\lambda = \sum_{n\in\mathbb N} \sigma_n^\nu\sigma_n^2/(\sigma_n^2 + \lambda) \langle w,v_n\rangle_{X} v_n = \sum_{n\in\mathbb N} \sigma_n^\nu \langle  w_\lambda,v_n\rangle_{X} v_n
> $$
> with $\lVert w_\lambda\rVert_{X}^2 \le \sup_{n} \sigma_n^2/(\sigma_n^2+\lambda) \lVert w\rVert_{X}^2 \le \lVert w\rVert_{X}^2$. Hence, $x_\lambda \in X_{\nu,\delta}$ whenever we assume $x^\dagger \in X_{\nu,\delta}$.
>
> ## Action:
> We will cite the mentioned paper and discuss the relation of the Lojasiewicz condition to linear IP.
>
> # Q3: Preventing explosion of SGD
> Thank you for your question. It is true that boundedness (bds) of the iterates of SGD, its necessity and its consequences is an active field of research. The recent work of Liu & Yuan (JMLR 2024) shows a.s. bds of vanilla SGD under convexity and the ABC-condition for step-sizes $\alpha_k = C_\alpha k^{-q}$ with $q \in (2/3,1)$, see Thm 13 therein. Using vanishing Tikhonov regularization, we can go as low as $q \in (1/2,1]$ for a.s. bds and $q \in (0,1]$ for bds in $L^2$. In the case $q \in (0,1/2)$ bds is especially difficult to achieve, since the corresponding martingale (the sum over all of the accumulated noise terms) does not converge.
>
> It is true that for convergence of the function value (or convergence to the set of global minima), bds of the trajectory is not necessarily required (see e.g. Louzi (2024) or Garrigos et al. (2025)). Still, there are certain benefits from having bounded (or convergent) trajectories, such as benefiting from averaging over the iterates for objective functions satisfying the PL-inequality, see e.g. Dereich & Kassing (EJP 2023). Bds can easily be achieved for coercive functions without a Tikhonov regularization (see e.g. Thm 1 in Mertikopoulos et al. (NeurIPS 2020), Lem D.1 in Dereich & Kassing (EJP 2023), or Lem 1.6 in Louzi (2024)). However, in this work we show that for general (possibly non-coercive) convex functions a vanishing Tikhonov regularization is strong enough to imply bds of the paths if the decay of the regularization parameter is not too fast.
>
> ## Action:
> We will include a paragraph with the above discussion.
>
> # Q4: ERM
> Thank you very much, we will add a comment.

---

> > ### Author Response · Authors · 2025-08-05
> >
> > Dear reviewer, we express once again our gratitude to your time and effort for the review. We have also put a lot of effort in the rebuttal and would appreciate a reaction (and a change of scores) if you agree with our improvements.

---

> > > ### Author Response · Authors · 2025-08-08
> > >
> > > Dear reviewer, the time for the rebuttal is running out. We would still be very interested in your opinion on our discussion of your questions. Best wishes from the authors

---

### Official Review · Reviewer_7hxJ · 2025-07-03

**Clarity:** 3
**Significance:** 2
**Originality:** 3
**Rating:** 4
**Confidence:** 4

**Summary:**

This paper analyzes the Tikhonov regularized stochastic gradient descent (reg-SGD) algorithm for minimizing convex and $L$-smooth functions in separable real Hilbert spaces.

Assuming a typical ABC-type condition on the noise sequence and without requiring any boundedness assumptions, the authors prove almost sure and $L^2$ convergence of the last iterates of reg-SGD to the minimum-norm solution. For polynomial stepsize and regularization schedules, they further establish sublinear convergence rates in both almost sure and $L^2$-convergence senses, with refined results under the \L ojasiewicz condition.

Numerical experiments are provided to support the theoretical results.

**Questions:**

Please see the weakness part and should address W2.

**Ethical Concerns:**

["NO or VERY MINOR ethics concerns only"]

**Final Justification:**

I have raised my score as I believe the new arguments are correct and appreciate the efforts made by the authors. However, given that the paper is 33 pages long and the modifications involve multiple changes to various theorems and lemmas, I am unable to verify all the details without reviewing the complete revision. As reviewers, we are not in a position to check every detail within the very limited time available.

**Limitations:**

yes

**Quality:**

3

**Strengths And Weaknesses:**

S1. This paper establishes almost sure and $L^2$ convergence of reg-SGD to the minimum-norm solution without imposing additional boundedness assumptions. Moreover, the almost sure boundedness of the iterates is proved, highlighting the stability of reg-SGD to vanilla SGD.

S2. This paper provides clear guidance on explicit choices of polynomial stepsize and regularization schedules that guarantee convergence of reg-SGD, enhancing the applicability of the analysis in pratical applications.

W1. Typos.
1) Line 228, 631, 742: It should be "$\Vert X_k - x_{\lambda_{k+1}}\Vert_{\mathcal{X}}^2$".
2) Line 666-667: It should be "$\lambda_k - \lambda_{k+1} = o(\alpha_k \lambda_k)$".
3) Line 701-702: It should be "$\cdots - \frac{p}{2\varepsilon^\prime C_\alpha}\Vert x_{*}\Vert_{\mathcal{X}}^2$".
4) Line 726, 750: It should be "$p+q\leq 1$".
5) Line 728: It should be "$p+q = \min\\{1+p-\beta,2q-\beta\\}$".
6) Line 749: It should be "$E[\varphi_k\ \vert F_{k-1}] \leq \cdots \varphi_{k-1} + \cdots$".

W2. Gaps in the Assumptions of Theorem 2.1. The claim in Line 658-659 that $\varphi_k\to 0$ almost surely via Corollary B.8 lacks sufficient justification. To apply Corollary B.8, it is necessary to verify that

$
1-\tau_k = \min\\{\alpha_k\lambda_k, 2\alpha_k\lambda_k(1-\frac{L+\lambda_k}{2}\alpha_k)-(\frac{L+\lambda_k}{2}A+A+2L+2\lambda_k)\alpha_k^2\\} \geq 0
$

holds on $B_\infty(N)$ for sufficiently large $k$. However, under the assumptions of Theorem 2.1, this inequality may not be satisfied. As a counterexample, define $C_1 \triangleq \frac{LA}{4} + \frac{A}{2} + L$, set $\lambda_k = \frac{1}{\sqrt{k}}$ for $k\in Z_{+}$, and let
$$
\alpha_k = \begin{cases}
\frac{2}{C_1}\lambda_{k}\quad &\mbox{if}\ k = 2^n\ \mbox{for some}\ n\in \mathbb{N}\\\\
\frac{\lambda_k}{\log k}\quad &\mbox{otherwise}.
\end{cases}
$$
Then it follows from
$
\sum_{n\geq 0} \frac{1}{2^n} < \infty,\quad \sum_{k\geq 3}\ \frac{1}{k(\log k)^2} < \infty\quad \mbox{and}\quad \sum_{k\geq 3}\ \frac{1}{k\log k} = + \infty,
$
that $\sum_k \alpha_k^2 < \infty$ and $\sum_k \alpha_k \lambda_k = \infty$, satisfying the requirements of Theorem 2.1. However, for any $k = 2^n$ with $n\in \mathbb{N}$, we have

$
1-\tau_k \leq 2\alpha_k \lambda_k - 2C_1 \alpha_k^2 = 2\alpha_k \lambda_k (1-C_1\frac{\alpha_k}{\lambda_k}) = \alpha_k \lambda_k (1-C_1 \cdot \frac{2}{C_1}) = -\alpha_k \lambda_k < 0,
$

which implies that $1-\tau_k < 0$ occurs infinitely many times. Consequently, the key condition for applying Corollary B.8 is violated, and the almost sure convergence of $\varphi_k$ cannot be guaranteed under the stated assumptions.

W3. Assumptions in Theorem 2.2 and proof argument in Line 668. See W2.

W4. Proof argument of Theorem 2.4. In Line 756, the derivation of $\Psi_k\in \mathcal{O}(k^{-1+q})$ by an argument analogous to the proof of Theorem C.5 is potentially problematic. When $q>0$ and $q = 1-p$, the proof of Theorem C.5 requires the condition that $2C_\lambda C_\alpha > 1-q$ (see Line 698-699), which may not be satisfied under the assumptions of Theorem 2.4 when $\beta < 1-q$.

W5. Proof argument of Theorem 2.4. In Line 758-759, the application of Corollary B.7 to establish almost sure convergence of $\{\varphi_k\}$ is not fully justified. The corollary requires a uniform lower bound on $\{\varphi_k\}$, but the almost sure boundedness of $\{\tilde{E}_k\}$ alone is insufficient, since $\varphi_k = \tilde{E}_k k^\beta$ with $\beta > 0$ by definition. This raises concerns about the applicability of the corollary in this context.

---

> ### Author Rebuttal · Authors · 2025-07-30
>
> Dear reviewer,
>
> We sincerely thank you for the time spent evaluating our work, in particular the in-depth checking of the proofs. We very much hope you will be awarded a best reviewer award. Your careful review helped us to improve clarity of the proofs for the reader. Thanks! All your points can be fixed with minor modifications. In the following, we will provide point by point responses to your comments. If you agree with the corrections we would appreciate if you would raise the score.
>
> To allow a full revision we give the full details how the corrected version will look like.
>
> ## W1.
> Thanks, we will correct the typos.
>
> ## W2/W3. Gaps in the Assumptions of Theorem 2.1 and Theorem 2.2.
>
> The problem you have identified here lies in our application of Corollary B.8 which in its current form does not allow a recursive inequality of form
> $$\mathbb E[Y_{k+1} \mid \mathcal F_k] \le (1+C_k - A_k) Y_k + B_k,$$
> where $\sum C_k<\infty, \sum B_k <\infty$ and $\sum A_k=\infty$ almost surely. However, such an extension is an immediate consequence of the Robbins-Siegmund theorem (see below). Having the corollary in this form, we can split $\tau_k$ on $\mathbb B_\infty(N)$ into
>     $$ \tau_k \le 1- \alpha_k\lambda_k + \Big(\frac{L+\lambda_{k}}{2} A +A+2L+2\lambda_k\Big) \alpha_{k}^2 =: 1-A_k + C_k, $$
> such that we can apply the updated Corollary B.8. For W3 and the concerns in the proof of Theorem 2.2, we similarly split $\tau_k$ into $1-A_k+C_k$, such that $\sum A_k=\infty$ and $\sum C_k <\infty$. Below are the full details and how the manuscript will be changed:
>
> ### Action 1 (Updated Corollary B.8):
>
> We will include the following extended Corollary 3.8:
>
> **Corollary:** Let $(\mathcal F_k)$ be a filtration and $(Y_k)$, $(A_k)$, $(B_k)$ and $(C_k)$ be non-negative and adapted processes satisfying almost surely that
> $$
> \sum_{k = 1}^\infty A_k = \infty \quad , \quad \sum_{k=1}^\infty B_k <\infty\quad \text{and}\quad \sum_{k=1}^\infty C_k<\infty.
> $$
>
> Moreover, suppose that for all $k \in \mathbb N$ one has almost surely that
> $$
> \mathbb E[Y_{k+1} \mid \mathcal F_k] \le (1+C_k-A_k)Y_k+B_k.
> $$
> Then $Y_k \to 0$ holds almost surely as $k \to \infty$.
>
> *Proof:* The proof follows the same lines as the proof of Lemma A.2 in [54]. For completeness, we provide the details. Compared to Lemma B.6, we have $Y_k = Y_k$, $X_k = A_kY_k$, $Z_k = B_k$ and $\gamma_k = C_k$. Using Lemma B.6 we obtain the existence of $Y_\infty$ which is almost surely finite, integrable and satisfies $Y_n\to Y_\infty$ almost surely. Additionally, we have that $\sum_{k=1}^\infty X_k = \sum A_k Y_k<\infty$ implying that $\liminf_{k\to\infty} Y_k = 0$, where we have used the assumption $\sum_{k=1}^\infty A_k = \infty$ almost surely. Since the limit inferior and limit coincide for converging sequences, the assertion follows by
> $$ Y_\infty = \lim_{k\to\infty} Y_k = \liminf_{k\to\infty} Y_k = 0\quad \text{almost surely}.$$
>
> ### Action 2 (Changes in Thm 2.1):
>
> Moreover, we have updated the the proof of Theorem 2.1 from line 655 as follows:
>
> Fix $N \in \mathbb N$ and for $k \ge N$ denote $\mathbb B_k(N)=  \lbrace  \alpha_i \le \frac{1}{L+\lambda_i} : i = N, \dots, k \rbrace $.
> Then, for all $ k > N$
> $$
> \mathbb E[1_{\mathbb B_k(N)}(\varphi_k + E_k) \mid \mathcal F_{k-1}] \le \tau_k 1_{\mathbb B_{k-1}(N)}(\varphi_{k-1}+E_{k-1}) + \alpha_k \lambda_k (\|x_\ast\|^2-\|x_{\lambda_k}\|^2) + \frac{\lambda_{k}-\lambda_{k+1}}{2}\|x_\ast\|^2  + \alpha_k^2\Bigl(A\frac {\lambda_{k}}2 \|x_\ast\|^2 +C \Bigr)\Bigl( 1+ \frac{L+\lambda_k}{2} \Bigr),
> $$
> where
> $$
> \tau_k =  \max\Big( 1-\alpha_k\lambda_k, 1-2\lambda_{k}\alpha_{k}\Big(1-\frac{L+\lambda_k}2\alpha_{k}\Big)+\Bigl(\frac{L+\lambda_{k}}{2} A +A+2L+2\lambda_k\Bigr) \alpha_{k}^2  \Big)
> $$
> and we have used that $\varphi_{k-1}+E_{k-1} \ge 0$ and $B_{k-1}(N) \supset B_k(N)$. On the event $B_{k-1}(N)$ we have
> $$
> \tau_k \le 1-\alpha_k\lambda_k + \Bigl(\frac{L+\lambda_{k}}{2} A +A+2L+2\lambda_k\Bigr) \alpha_{k}^2 =: 1-A_k + C_k \,,
> $$
> where by assumption $\sum_{k\in\mathbb N} C_k <\infty$ and $\sum_{k\in\mathbb N} A_k = \infty$ almost surely. Now, we can apply Corollary B.8 for the process $(1_{\mathbb B_k(N)} (\varphi_k+E_k), k\ge N)$
> to deduce that, on $B_\infty(N) = \bigcap_{k \ge N} B_k(N)$, one has $\varphi_k \to 0$ almost surely as $k \to \infty$. Since $\alpha_k \to 0$ almost surely one has
> $$
> \mathbb P \Big( \bigcup_{N \in \mathbb N} B_\infty(N) \Big)=1
> $$
> and, thus, the proof of the theorem is finished.
>
> ### Action 3 (Changes in Thm 2.2):
>
> The beginning of the proof of Theorem 2.2 now reads as follows:
>
> First, we prove the theorem assuming that (15) holds. By assumption, one has $\sum_{k\in\mathbb N} A_k = \infty$, $\sum_{k\in\mathbb N} C_k <\infty$, $\sum_{k \in \mathbb N} \alpha_k^2<\infty$, $\sum_{k \in \mathbb N} \alpha_k\lambda_k (\|x_\ast\|^2-\|x_{\lambda_k}\|^2)<\infty$ and $\sum_{k = 1 }^\infty(\lambda_k-\lambda_{k+1}) = \lambda_1 < \infty$. Therefore, after taking expectations in (18), we can apply Lemma B.5 to deduce that $\mathbb E[\varphi_k]\to 0$ and $\mathbb E[E_k] \to 0$.
>
> ## W4. Proof argument of Theorem 2.4. (Line 756)
>
> Thanks for noticing this issue. In view of inequality (22) in Theorem C.5, we need the constant $2C_\lambda C_\alpha$ to be strictly larger than the exponent in the convergence rate if $q=1-p$. For proving Theorem C.7 it is sufficient to show that $\Psi_k \in \mathcal O(k^{-\min(\beta,1-q)})$. Thus, we only require $2C_\lambda C_\alpha > \min(\beta,1-q)$.
> In Theorem 2.4 in the main text, we excluded the case $q=1-p$ to streamline presentation.
>
> ### Action 4 (Assumptions in Theorem 2.4):
>
> In Theorem C.7 we will assume that $2C_\lambda C_\alpha > \min(\beta,1-q)$. In the proof of Theorem C.7 we will write that $\Psi_k \in \mathcal O(k^{-\min(\beta,1-q)})$.
>
> ## W5. Proof argument of Theorem 2.4. (Line 758-759)
>
> Thank you very much for noticing this. We can use the idea from our response to W4 to show that $\tilde E_k k^{\min(\beta, 1-q)}$ is uniformly bounded from below. Then, we can apply Corollary B.7 to show that $\tilde E_k \in \mathcal O(k^{-\min(\beta,1-q)})$ almost surely, which is sufficient for proving Theorem C.7.
>
> ### Action 5 (Proof adaption of Theorem 2.4):
>
> We define $\beta \in (0,2q-1)$, $\tilde \beta = \min(\beta, 1-q)$ and $\varphi_k = \tilde E_k k^{\tilde \beta}$. The end of the proof now reads:
>
> Therefore, analogously to the proof in the deterministic setting, we get $\Psi_k \in \mathcal O(k^{-\tilde \beta})$, i.e. $(\Psi_k k^{\tilde \beta}, k \in \mathbb N)$ is bounded and, subsequently, $(\varphi_k, k \in \mathbb N)$ is uniformly bounded from below.
>         Now, since $\beta < 2q-1$ we get $\sum k^{-2q+\beta} < \infty$. Hence, we can apply Corollary~B.7 to get almost sure convergence of $(\varphi_k)_{k \in \mathbb N}$ and, thus,
> 		$
> 			\tilde E_k \in \mathcal O(k^{-\tilde \beta})
> 		$
>         almost surely.
>         Together with $\tilde E_k = E_k -\Psi_k$ and $\Psi_k = \mathcal O(k^{-\tilde \beta})$, this implies that
> 		$
> 			E_k \in \mathcal O( k^{-\tilde \beta})
> 		$
>         almost surely.

---

> > ### Author Response · Authors · 2025-08-05
> >
> > Dear reviewer, we express once again our gratitude to your time and effort for the review. We have also put a lot of effort in the rebuttal and would appreciate a reaction (and a change of scores) if you agree with our improvements.

---

### Note · Authors · 2025-08-12

Since the rebuttal did not go entirely according to the NeurIPS guidelines, we decided to quickly summarize. (i) We had a number of minor comments (thanks!) that we addressed. (ii) We had requests for more literature review, which we addressed. In particular, we highlight that similar conditions/rates appeared earlier in the literature for **deterministic constrained** optimization with increasing penalty parameter. (iii) We had a very careful review (thanks!) highlighting three minor problems in the proofs of the appendix. Those where easy to fix. We want to highlight: Our rebuttal gives a **full revision of the few corrected arguments**, that should be easy to verify. Every change that we make to the paper is spelled out in the chat box. The reviewer acknowledge the changes look correct to him, but without carefully checking. Note: Since all details are presented in the chat box (and easy to check given the depth of the original review), this is fully in the scope of NeurIPS rebuttals. This is not a reason to suggest a complete make-over and resubmission elsewhere.

---

### Decision · Program_Chairs · 2025-09-17

**Decision:**

Accept (poster)

**Comment:**

**Summary:**
This paper investigates the property of the Tikhonov regularized stochastic gradient descent (reg-SGD) algorithm for solving convex and $L$-smooth objectives in separable real Hilbert space. Under an ABC-type assumptions on the stochastic noise, convergence of the last iterates of reg-SGD is presented. For polynomial stepsize and regularization schedules, theoretical guarantees are also provided. The theoretical result is supported with numerical experiments.


**Strength:**
 - A well presented paper in terms of written, clarity.
 - Important contributions to the understand of SGD algorithm.
 - Weak assumptions, only requires an ABC-type assumption and does not need bounded gradient assumption.

**Weakness:**
 - The organization of the paper can be further improved.
 - Missing discussion on connections with existing literature.
 - The first submission contains mathematical flaws.

**Discussion summary:**
The paper provides a solid theoretical analysis of SGD with Tikhonov regularization, and the findings make an important contribution to understanding the method. During the rebuttal phase, the reviewers’ concerns were satisfactorily addressed. The authors are advised to revise the manuscript carefully, especially given that the initial submission contained mathematical errors.